# Residual Q-Learning: Offline and Online Policy Customization without Value

**Chenran Li**[1]*, **Chen Tang**[1]*, **Haruki Nishimura**[2], **Jean Mercat**[2],
**Masayoshi Tomizuka**[1], **Wei Zhan**[1]
[1] University of California Berkeley, [2] Toyota Research Institute, USA
{chenran_li,chen_tang,tomizuka,wzhan}@berkeley.edu
{haruki.nishimura,jean.mercat}@tri.global

## Abstract

Imitation Learning (IL) is a widely used framework for learning imitative behavior from demonstrations. It is especially appealing for solving complex real-world tasks where handcrafting reward function is difficult, or when the goal is to mimic human expert behavior. However, the learned imitative policy can only follow the behavior in the demonstration. When applying the imitative policy, we may need to *customize* the policy behavior to meet different requirements coming from diverse downstream tasks. Meanwhile, we still want the customized policy to maintain its imitative nature. To this end, we formulate a new problem setting called *policy customization*. It defines the learning task as training a policy that inherits the characteristics of the prior policy while satisfying some additional requirements imposed by a target downstream task. We propose a novel and principled approach to interpret and determine the trade-off between the two task objectives. Specifically, we formulate the customization problem as a Markov Decision Process (MDP) with a reward function that combines 1) the inherent reward of the demonstration; and 2) the add-on reward specified by the downstream task. We propose a novel framework, *Residual Q-learning (RQL)*, which can solve the formulated MDP by leveraging the prior policy *without knowing the inherent reward or value function of the prior policy*. We derive a family of residual Q-learning algorithms that can realize offline and online policy customization, and show that the proposed algorithms can effectively accomplish policy customization tasks in various environments. Demo videos and code are available on our website: https://sites.google.com/view/residualq-learning.

## 1 Introduction

Imitation learning (IL) is a widely used framework for learning imitative behavior from demonstrations. It is especially appealing for solving complicated real-world tasks where handcrafting reward function is difficult, or when the goal is to mimic expert behavior. For instance, IL has been widely studied for synthesizing human-like agents from large-scale human demonstrations on challenging real-world tasks, such as autonomous driving [4, 7, 2] and robot manipulation [59, 37, 39]. However, when applying the imitative policy in open real-world scenarios, we may encounter different circumstances that impose diverse requirements beyond merely following the behavior in the demonstration, such as reaching particular goals [45], enforcing safety [36], or complying with certain behavior preferences [61]. In this case, we need to *customize* the imitative policy to meet those task-specific objectives. Meanwhile, in many applications, we may still want the customized policy to possess some inherent characteristics of the expert demonstration (e.g., human-like).

---

*Equal Contribution

37th Conference on Neural Information Processing Systems (NeurIPS 2023).

To this end, we formulate a new problem setting we refer to as *policy customization*. In the policy customization problem, we are given a pre-trained policy, and the task is to train a new policy that 1) inherits the properties of the prior policy; 2) satisfies some additional requirements imposed by a given downstream task. At first glance, this newly formulated problem may seem to be seamlessly solved by existing reinforcement learning (RL) fine-tuning algorithms [44, 40, 61, 51], where RL is applied to fine-tune a pre-trained policy obtained from behavior cloning (BC) or offline RL [31, 29, 34]. In particular, some RL fine-tuning methods combine different forms of IL objectives with RL to accelerate policy training, e.g., maximizing the policy likelihood on demonstration [44], regularizing the KL-divergence between the fine-tuned and prior policies [40, 61]. Different from the policy customization problem we consider, the IL objectives are introduced heuristically as regularization in RL fine-tuning. Solving policy customization instead requires a *principled* way to design the learning objective to jointly optimize the policy's performance on the imitative and downstream tasks, which has not been well-studied in the RL fine-tuning literature.

In this work, we propose such a principled approach to interpret and design the policy customization objective with a solid theoretical basis. Formally, we formulate policy customization as a Markov decision process (MDP) whose reward function is a linear combination of 1) the inherent reward of the demonstration; and 2) an add-on reward specified by the downstream task. The reward weight directly determines the trade-off between the imitative and downstream tasks for policy optimization. However, we cannot solve this MDP with existing RL algorithms, since we can only access the imitative policy without knowing its underlying reward.

To tackle this challenge, we propose a novel *Residual Q-Learning (RQL)* framework that can solve the formulated MDP *without knowing the inherent reward or value function of the prior policy*. Specifically, we define a residual Q-function that, together with the log-likelihood of the prior policy, can conveniently construct the maximum-entropy policy solving the target MDP. We show that the residual Q-function can be iteratively learned with an update rule derived from soft Bellman equation [17]. Hence, the proposed RQL method can be applied on top of any value-based maximum-entropy RL algorithms. In particular, we derive residual soft Q-learning and residual soft actor-critic—two model-free policy customization algorithms through RL fine-tuning—which are adapted from soft Q-learning [17] and soft actor-critic [18] respectively. In addition, we introduce a model-based policy customization algorithm adapted from maximum-entropy Monte Carlo Tree Search (MCTS) [55], which enables *zero-shot online* policy customization when a dynamics model is available. In our experiments, we show that the proposed RQL algorithms can effectively customize the policies toward the additional task objective while maintaining their performance on the basic tasks for which the prior policies are trained.

## 2 Policy Customization with Residual Q-Learning

### 2.1 Problem Formulation: Policy Customization

We consider a MDP defined by the tuple $\mathcal{M} = (\mathcal{S}, \mathcal{A}, r, p)$, where $\mathcal{S}$ is the state space, $\mathcal{A}$ is the action space, and $r : \mathcal{S} \times \mathcal{A} \to \mathbb{R}$ is the reward function. Without loss of generality, we introduce the formulation under continuous state and action spaces. The state transition probability function $p : \mathcal{S} \times \mathcal{A} \times \mathcal{S} \to [0, \infty)$ represents the probability density of the next state $s_{t+1} \in \mathcal{S}$ given the current state $s_t \in \mathcal{S}$ and action $a_t \in \mathcal{A}$. While the following formulation applies whether the reward function $r$ is given or not, we are primarily interested in the case when $r$ remains *unknown* and, instead, expert demonstrations are available to indicate desirable behaviors. We assume that we have access to a prior policy $\pi : \mathcal{S} \times \mathcal{A} \to [0, \infty)$ pre-trained with imitation learning [15, 22] methods. In particular, we follow the common practice and model $\pi$ as a maximum-entropy policy to account for sub-optimal demonstration [60], which is especially important for human demonstration. Specifically, the policy $\pi$ maximizes the entropy-augmented cumulative reward that is inherent in the demonstrations. The resulting optimal policy follows a Boltzmann distribution [17] as follows:

$$\pi\left(\boldsymbol{a}|\boldsymbol{s}\right) = \frac{1}{Z_s} \exp\left(\frac{1}{\alpha} Q^*(\boldsymbol{s}, \boldsymbol{a})\right), \tag{1}$$

where $Q^*(\boldsymbol{s}, \boldsymbol{a})$ is the soft Q-function as defined in [17] and it satisfies the soft Bellman equation:

$$Q^*(\boldsymbol{s}, \boldsymbol{a}) = r(\boldsymbol{s}, \boldsymbol{a}) + \gamma \mathbb{E}_{\boldsymbol{s}' \sim p(\cdot|\boldsymbol{s}, \boldsymbol{a})} \left[\alpha \log \int_{\mathcal{A}} \exp\left(\frac{1}{\alpha} Q^*(\boldsymbol{s}', \boldsymbol{a}')\right) d\boldsymbol{a}'\right], \tag{2}$$

and $Z_s$ is the normalization factor defined as $\int_{\mathcal{A}} \exp\left(\frac{1}{\alpha} Q^*(s, a)\right) da$. The temperature coefficient $\alpha$ determines the relative importance of entropy and reward in the maximum-entropy policy [17]. It is straightforward to derive from Eqn. (1) that:

$$Q^*(s, a) = \alpha \log \pi(a|s) + \alpha \log Z_s. \tag{3}$$

Instead of directly deploying the policy $\pi$, we aim to *customize* it to satisfy some additional requirements for a given downstream task, specified by an add-on reward function $r_R : \mathcal{S} \times \mathcal{A} \to \mathbb{R}$. Meanwhile, we want the customized policy to inherit the characteristics of the original policy. Formally, we aim to leverage the pre-trained prior policy $\pi$ to train a new policy for the MDP defined by $\hat{\mathcal{M}} = (\mathcal{S}, \mathcal{A}, \omega r + r_R, p)$. The new reward function is defined as a weighted sum of the basic reward $r$ and the add-on reward $r_R$ so that the customized policy meets both requirements. Under this setting, the main challenge lies in how to find the optimal policy for the new task without knowing the prior reward $r$ but only the imitative policy $\pi$.

## 2.2 Residual Q-Learning

Since the reward $r$ is unknown, we cannot directly train the new policy to maximize the total reward through RL. One plausible solution is to learn the inherent reward via inverse RL [12] and then combine the inferred reward with the add-on reward for RL training. However, practical inverse RL algorithms require jointly optimizing the policy and reward function, which could be challenging to yield good performance in certain environments when compared to imitation learning [16]. Instead, we aim to explore a solution without the need to explicitly infer the underlying reward to derive a more flexible policy customization framework. It turns out that we can leverage the imitative policy $\pi$ to construct the new policy *without knowing $\pi$'s underlying reward or value function*. The proposed method can be applied on top of any *value-based maximum-entropy* RL algorithms [17, 18, 55], which rely on the soft Bellman update operator defined as:

$$\hat{Q}_{t+1}(s, a) = r_R(s, a) + \omega r(s, a) + \gamma \mathbb{E}_{s' \sim p(\cdot|s, a)} \left[ \hat{\alpha} \log \int_{\mathcal{A}} \exp\left(\frac{1}{\hat{\alpha}} \hat{Q}_t(s', a')\right) da' \right], \tag{4}$$

where $\hat{Q}_t$ is the estimated soft Q-function for $\hat{\mathcal{M}}$ at the $t^{\text{th}}$ iteration. Instead of directly applying the soft Bellman update operator to iterate $\hat{Q}_t$ during policy learning, we define a *residual* Q-function as $Q_{R,t} := \hat{Q}_t - \omega Q^*$ and attempt to iterate this residual Q-function during policy learning. We then derive an update rule for $Q_{R,t}$ that does not depend on the reward or value function of $\pi$:

$$Q_{R,t+1}(s, a)$$

$$= r_R(s, a) + \omega r(s, a) + \gamma \mathbb{E}_{s'} \left[ \hat{\alpha} \log \int_{\mathcal{A}} \exp\left(\frac{1}{\hat{\alpha}} \hat{Q}_t(s', a')\right) da' \right] - \omega Q^*(s, a), \tag{5a}$$

$$= r_R(s, a) + \omega Q^*(s, a) - \omega \gamma \mathbb{E}_{s'} \left[ \alpha \log \int_{\mathcal{A}} \exp\left(\frac{1}{\alpha} Q^*(s', a')\right) da' \right]$$
$$+ \gamma \mathbb{E}_{s'} \left[ \hat{\alpha} \log \int_{\mathcal{A}} \exp\left(\frac{1}{\hat{\alpha}} \hat{Q}_t(s', a')\right) da' \right] - \omega Q^*(s, a), \tag{5b}$$

$$= r_R(s, a) - \omega \alpha \gamma \mathbb{E}_{s'} \log Z_{s'}$$
$$+ \gamma \mathbb{E}_{s'} \left[ \hat{\alpha} \log \int_{\mathcal{A}} \exp\left(\frac{1}{\hat{\alpha}} (Q_{R,t}(s', a') + \omega \alpha \log \pi(a'|s') + \omega \alpha \log Z_{s'})\right) da' \right], \tag{5c}$$

$$= r_R(s, a) - \omega \alpha \gamma \mathbb{E}_{s'} \log Z_{s'} + \omega \alpha \gamma \mathbb{E}_{s'} \log Z_{s'}$$
$$+ \gamma \mathbb{E}_{s'} \left[ \hat{\alpha} \log \int_{\mathcal{A}} \exp\left(\frac{1}{\hat{\alpha}} (Q_{R,t}(s', a') + \omega \alpha \log \pi(a'|s'))\right) da' \right], \tag{5d}$$

$$= r_R(s, a) + \gamma \mathbb{E}_{s'} \left[ \hat{\alpha} \log \int_{\mathcal{A}} \exp\left(\frac{1}{\hat{\alpha}} (Q_{R,t}(s', a') + \omega \alpha \log \pi(a'|s'))\right) da' \right], \tag{5e}$$

where Eqn. (5b) results from expressing the reward $r$ with the soft Q-function $Q^*$ based on Eqn. (2). Eqn. (5c) results from the definition of the residual Q-function and Eqn. (3). Please refer to Appendix A for a step-by-step breakdown of the derivation process. In each iteration, we can then define the policy corresponding to the current estimated $\hat{Q}_t$ without computing $\hat{Q}_t$:

$$\hat{\pi}_t(a|s) \propto \exp\left(\frac{1}{\hat{\alpha}} (Q_{R,t}(s, a) + \omega \alpha \log \pi(a|s))\right). \tag{6}$$

One remaining issue is that we do not have access to the underlying temperature coefficient $\alpha$ of the prior policy $\pi$. Note that $\alpha$ is multiplied upon the reward weight $\omega$ in Eqn. (5) and Eqn. (6). In practice, the reward weight is a hyperparameter to tune. Hence, we can define $\omega' = \omega\alpha$ and directly treat $\omega'$ as the hyperparameter without the need to know the true $\alpha$.

## 3  Practical Algorithms

In this section, we present three practical algorithms for policy customization under different settings based on the update rule of residual Q-function in Eqn. (5) and the maximum-entropy policy defined in Eqn. (6). In these algorithms, the goal is to find a policy that solves the MDP $\hat{\mathcal{M}}$ given a prior policy $\pi$ and the add-on reward function $r_R$. We first introduce two model-free algorithms to learn the residual Q-function and customize the policy through additional RL steps, which are adapted from soft Q-learning [17] and soft actor-critic [18] respectively. Then we introduce a model-based policy customization algorithm adapted from maximum-entropy MCTS [55], which enables *zero-shot online* policy customization when a dynamics model is available.

**Residual Soft Q-Learning.** We adapt the soft Q-learning [17] algorithm to a residual soft Q-learning algorithm for offline policy customization (i.e., policy customization via additional RL training steps) when the action space is discrete. The residual algorithm is essentially the same as the soft Q-learning. The only difference is that we use a function approximator parameterized by $\theta$ to model the residual Q-function instead of the soft Q-function and denote it as $Q_{R,\theta}(\boldsymbol{s}, \boldsymbol{a})$. At each iteration, the parameters are updated by taking gradient steps to minimize the temporal-difference (TD) error:

$$J_{Q_R}(\theta) = \mathbb{E}_{(\boldsymbol{s}_t, \boldsymbol{a}_t, r_{R,t}, \boldsymbol{s}_{t+1}) \sim \mathcal{D}} \left[ \frac{1}{2} \left( Q_{R,\bar{\theta}}^{\text{target}}(\boldsymbol{s}_t, \boldsymbol{a}_t) - Q_{R,\theta}(\boldsymbol{s}_t, \boldsymbol{a}_t) \right)^2 \right], \tag{7}$$

where $\mathcal{D}$ is the replay buffer and the target residual Q-value $Q_{R,\bar{\theta}}^{\text{target}}(\boldsymbol{s}_t, \boldsymbol{a}_t)$ is computed as:

$$Q_{R,\bar{\theta}}^{\text{target}}(\boldsymbol{s}_t, \boldsymbol{a}_t) = r_{R,t} + \gamma\hat{\alpha} \log \sum_{\boldsymbol{a}' \in \mathcal{A}} \exp \left( \frac{1}{\hat{\alpha}} \left( Q_{R,\bar{\theta}}(\boldsymbol{s}_{t+1}, \boldsymbol{a}') + \omega' \log \pi(\boldsymbol{a}'|\boldsymbol{s}_{t+1}) \right) \right), \tag{8}$$

with $\bar{\theta}$ being the target parameters. The above objective function can be derived straightforwardly from the original objective function of soft Q-learning. We provide the detailed derivation in Appendix B for completeness. With a discrete action space, we can compute the policy distribution exactly given the residual Q-function and the prior policy as in Eqn. (6).

**Residual Soft Actor-Critic.** For tasks with continuous action spaces, we adapt the soft actor-critic [18] algorithm to a residual soft actor-critic algorithm. We aim to train a parameterized residual Q-function $Q_{R,\theta}(\boldsymbol{s}, \boldsymbol{a})$ and a policy network $\hat{\pi}_\phi(\boldsymbol{a}|\boldsymbol{s})$ by alternating between policy evaluation and improvement. In the policy evaluation step, we update the residual Q-function for evaluating the current policy. The loss function is essentially the one in Eqn. (7) but with the target residual Q-value computed with respect to the current policy $\hat{\pi}_\phi$:

$$Q_{R,\bar{\theta}}^{\text{target}}(\boldsymbol{s}_t, \boldsymbol{a}_t) = r_{R,t} + \gamma\mathbb{E}_{\boldsymbol{a}' \sim \hat{\pi}_\phi} \left[ Q_{R,\bar{\theta}}(\boldsymbol{s}_{t+1}, \boldsymbol{a}') + \omega' \log \pi(\boldsymbol{a}'|\boldsymbol{s}_{t+1}) - \hat{\alpha} \log \hat{\pi}_\phi(\boldsymbol{a}'|\boldsymbol{s}_{t+1}) \right] \tag{9}$$

The formula for the residual Q-value is derived from the modified Bellman update operator of soft actor-critic [18] following a similar procedure as in Eqn. (5). The detailed derivation is attached to Appendix C for completeness. In the policy improvement step, we improve the policy $\hat{\pi}_\phi$ leveraging the current estimated residual Q-value $Q_{R,\theta}(\boldsymbol{s}, \boldsymbol{a})$ and the prior policy $\pi$ by minimizing the expected KL-divergence between $\hat{\pi}_\phi$ and the desired maximum-entropy policy given the current $Q_{R,\theta}(\boldsymbol{s}, \boldsymbol{a})$, which is equivalent to minimizing the loss function below:

$$J_{\hat{\pi}}(\phi) = \mathbb{E}_{\boldsymbol{s}_t \sim \mathcal{D}} \left[ \mathbb{E}_{\boldsymbol{a} \sim \hat{\pi}_\phi} \left[ \hat{\alpha} \log \hat{\pi}_\phi(\boldsymbol{a}|\boldsymbol{s}_t) - Q_{R,\theta}(\boldsymbol{s}_t, \boldsymbol{a}) - \omega' \log \pi(\boldsymbol{a}|\boldsymbol{s}_t) \right] \right]. \tag{10}$$

In our experiments, we implement the algorithm based on the practical soft actor-critic algorithm presented in [19], which incorporates auto entropy adjustment and double residual Q-functions [14].

**Residual Maximum-Entropy MCTS.** The last algorithm we introduce is a model-based policy customization algorithm adapted from the maximum-entropy MCTS [55] algorithm, which we refer to as residual maximum-entropy MCTS. Our primary interest is in applying this algorithm in the online setting with an offline learned dynamics model. MCTS essentially solves an online planning

problem leveraging the prior policy and add-on reward function, which enables *zero-shot online policy customization*. Such a paradigm is particularly useful for problem domains like autonomous driving, where large-scale human driving datasets [11, 58, 53] are available for learning good imitative policies [1, 46] as well as accurate traffic predictive models for online planning [10, 33]. Note that MCTS-based methods inherently assume a discrete action space. In practice, we may extend it to tasks with continuous action space by discretizing the action space [47, 57].

To solve the online planning problem, maximum-entropy MCTS constructs a look-ahead tree $\mathcal{T}$ incrementally via Monte Carlo simulation. In the tree $\mathcal{T}$, each node $n(\boldsymbol{s}) \in \mathcal{T}$ corresponds to a state $s$. For each action $a \in \mathcal{A}$, the node $n(\boldsymbol{s})$ stores an estimated soft Q-value $\hat{Q}(\boldsymbol{s}, \boldsymbol{a})$ and a visit count $N(\boldsymbol{s}, \boldsymbol{a})$. At each iteration, simulations are conducted starting from the root node of the search tree, which consists of two stages: 1) querying a *tree policy* to select actions and traversing the tree until a leaf node is reached; 2) querying an *evaluation function* (e.g., Monte Carlo simulations leveraging a roll-out policy) at the leaf node to estimate the return-to-go. After simulation, the estimated returns are propagated backward to update all the nodes along the simulated paths. Afterward, the tree is grown by expanding the leaf nodes reached during the simulations. Different from a typical MCTS algorithm [8], maximum-entropy MCTS exploits maximum-entropy policy optimization. Hence, the tree policy used in maximum-entropy MCTS is defined as:

$$\hat{\pi}(\boldsymbol{a}|\boldsymbol{s}) = (1 - \lambda_s)\frac{\exp(\frac{1}{\hat{\alpha}}\hat{Q}(\boldsymbol{s}, \boldsymbol{a}))}{\sum_{\boldsymbol{a}' \in \mathcal{A}} \exp(\frac{1}{\hat{\alpha}}\hat{Q}(\boldsymbol{s}, \boldsymbol{a}))} + \lambda_s \frac{1}{|\mathcal{A}|}, \tag{11}$$

where $\lambda_s = \epsilon|\mathcal{A}|/\log(\sum_{\boldsymbol{a}} N(\boldsymbol{s}, \boldsymbol{a}) + 1)$ with $\epsilon$ as a hyperparameter. Given a simulated trajectory from the root node to a leaf node, i.e., $\{\boldsymbol{s}_0, \boldsymbol{a}_0, \cdots, \boldsymbol{s}_T, \boldsymbol{a}_T\}$, the Q-values of all the visited nodes are updated with a *softmax backup*:

$$\hat{Q}(\boldsymbol{s}_t, \boldsymbol{a}_t) = \begin{cases} r_R(\boldsymbol{s}_t, \boldsymbol{a}_t) + \omega r(\boldsymbol{s}_t, \boldsymbol{a}_t) + \gamma\hat{R} & t = T - 1, \\ r_R(\boldsymbol{s}_t, \boldsymbol{a}_t) + \omega r(\boldsymbol{s}_t, \boldsymbol{a}_t) + \gamma\hat{\alpha} \log \sum_{\boldsymbol{a}' \in \mathcal{A}} \exp\left(\frac{1}{\hat{\alpha}}\hat{Q}(\boldsymbol{s}_{t+1}, \boldsymbol{a}')\right) & t < T - 1, \end{cases} \tag{12}$$

where $\hat{R}$ is the return-to-go estimated by querying an evaluation function on the leaf node $n(\boldsymbol{s}_T)$. To adapt the algorithm to residual maximum-entropy MCTS, we instead have each node store an estimated residual Q-value $Q_R(\boldsymbol{s}, \boldsymbol{a})$. The tree policy is revised as:

$$\hat{\pi}(\boldsymbol{a}|\boldsymbol{s}) = (1 - \lambda_s)\frac{\exp(\frac{1}{\hat{\alpha}}\left(Q_R(\boldsymbol{s}, \boldsymbol{a}) + \omega' \log \pi(\boldsymbol{a}|\boldsymbol{s})\right))}{\sum_{\boldsymbol{a}' \in \mathcal{A}} \exp(\frac{1}{\hat{\alpha}}\left(Q_R(\boldsymbol{s}, \boldsymbol{a}) + \omega' \log \pi(\boldsymbol{a}|\boldsymbol{s})\right))} + \lambda_s \frac{1}{|\mathcal{A}|}. \tag{13}$$

During backward propagation, the residual Q-value is updated as follows:

$$Q_R(\boldsymbol{s}_t, \boldsymbol{a}_t) = \begin{cases} r_R(\boldsymbol{s}_t, \boldsymbol{a}_t) + \gamma R & t = T - 1, \\ r_R(\boldsymbol{s}_t, \boldsymbol{a}_t) + \gamma V_R(\boldsymbol{s}_{t+1}) & t < T - 1, \end{cases} \tag{14}$$

with $V_R(\boldsymbol{s}_{t+1}) = \hat{\alpha} \log \sum_{\boldsymbol{a}' \in \mathcal{A}} \exp\left(\frac{1}{\hat{\alpha}}\left(Q_R(\boldsymbol{s}_{t+1}, \boldsymbol{a}') + \omega' \log \pi(\boldsymbol{a}'|\boldsymbol{s}_{t+1})\right)\right)$. In particular, we use Monte Carlo simulations to estimate the return-to-go $R$. The roll-out policy is set as the prior policy $\pi$, and $R$ is computed as the cumulative return of the add-on reward $r_R$ over the roll-out trajectories. While the estimated $R$ is biased, it serves as a heuristic to guide the tree search to select nodes with future trajectories collecting high add-on rewards while staying close to the prior policy. The effect of this biased estimation will gradually be mitigated as the search tree expands to the terminal time step.

## 4 Related Work

Our residual Q-learning framework can be considered a principled approach to combine the objectives of IL and RL. Prior works have proposed different approaches for this purpose. In the IL literature, reward augmentation [9] has been introduced as a general framework to provide additional incentives from prior knowledge into the imitation learning process. Under the Generative Adversarial Imitation Learning (GAIL) framework, a surrogate reward can be conveniently incorporated to augment the learned discriminator for policy optimization [35, 3]. The augmented imitative reward is essentially the same as the total reward in our framework. The difference is that ours aims to further customize a trained imitative policy for downstream tasks, instead of regularizing IL with prior knowledge.

There are also a variety of works that directly design the objective function as a weighted mixture of IL and RL objectives. In some methods, like DQfD [21], a supervised classification loss is combined

with a TD loss to train the Q-function. It regularizes the learned Q-function to assign higher values to demonstrated actions. Other methods, such as DAPG [44] and BC-SAC [36], use a weighted mixture of behavior cloning loss and RL objectives in their loss function. SHAIL [25] formulates the imitation problem as a constrained RL problem with a constraint on the lower bound of policy likelihood on demonstrations. These approaches have shown that IL can help accelerate RL learning, especially under sparse rewards [21, 44], and vice versa [25]. As discussed in Sec. 1, similar objectives are also adopted in some RL fine-tuning algorithms [44, 40, 61, 51]. The pre-trained policy is not only used to initialize RL but also leveraged consistently over the fine-tuning stage as regularization. The introduction of RL has also been shown to effectively robustify behavior cloning [36] for challenging and safety-critical autonomous driving tasks. Conversely, IL regularization is used in offline RL approaches, such as TD3+BC [13] and CQL [31], to prevent overestimating the values on out-of-distribution actions and improve the robustness of offline RL policies. Different from these works, where the additional objective is treated as regularization, our framework formulates the learning task as a new MDP and provides a principled way to interpret and design the combined objective.

TS2C [56] is similar to ours from the perspective of having an imperfect expert—the prior policy in RQL can be considered as an imperfect expert for the overall task—and fusing knowledge extracted from this imperfect expert and RL exploration. The difference is that TS2C aims to train a policy towards a known reward; Thus, a value function can be trained to assert if the expert is reliable. In our case, since we rely on the prior policy to encode the basic task reward implicitly, we cannot use such a value function to govern the fusion between imitation and RL. One work in the literature that aligns with our motivation behind MCTS online customization is Deep Imitative Model [45], which aims to direct an imitative policy to arbitrary goals during online usage. It is formulated as a planning problem solved online to find the trajectory that maximizes its posterior likelihood conditioned on the goal. In contrast, we formulate the additional requirements as rewards instead of goals, which are more flexible to specify. Also, RQL is a unified framework for both offline and online customization.

In curriculum RL, the concept of learning the residuals of Q-functions has been adopted in boosted curriculum RL (BCRL) [28, 24]. Under the BCRL framework, a task curriculum is constructed by decomposing a complex task into a sequence of subtasks with increasing difficulty. For each task of the curriculum, its Q-function is modeled and learned as the sum of residuals, with each residual corresponding to a previous task of the curriculum, which was shown to outperform fitting a single function approximator [50, 28]. Similar to RL fine-tuning, BCRL focuses on expediting RL learning on a target task with the Q-functions learned on the previous tasks. The residual is learned to construct the Q-function for the target task with a *known* reward. Conversely, RQL aims at learning the residual to construct the Q-function for a new task whose reward function combines the *unknown* underlying reward of the expert and the *known* add-on reward.

## 5 Experiments

We evaluate the proposed algorithms on four environments selected from different domains: *CartPole* and *Continuous Mountain Car* environments from the OpenAI gym classic control suite [5], and *Highway* and *Parking* from the *highway-env* environments [32]. In our experiments, we implemented our algorithms upon Stable-Baselines3 [43] and its imitation library [16]. In Sec. 5.1, we provide the configurations of our experiments, including the settings of policy customization tasks in different environments, baselines, and evaluation metrics. In Sec. 5.2, we present and analyze the experimental results of RL offline policy customization. In Sec. 5.3, we demonstrate an example of zero-shot online customization leveraging the residual maximum-entropy MCTS algorithm. In Sec. 5.4, we investigate two representative RL fine-tuning methods under the context of policy customization and compare them with the proposed algorithms. Please refer to the appendices for detailed experiment configurations, implementation details, ablation studies, and additional experiments in MuJoCo [49].

### 5.1 Experiment Setup

**Environments.** The detailed configurations of the environments can be found in Appendix D. We mainly describe the policy customization task settings here. In each environment, we design the basic and add-on rewards to illustrate a practical application scenario, where the basic reward corresponds to some basic task requirements and the add-on reward reflects customized specifications such as behavior preference or additional task objectives.

- **CartPole balancing:** In the cartpole environment, the basic task is to balance the pole as long as possible. During policy customization, we demand an additional task that requires the cart to stay at the center of the rack. To validate if the additional objective is satisfied, we monitor the average absolute position error of the cart which is defined as $\bar{e}_x = \sum_{t=1}^{T} |x_t|/T$, where $x_t$ is the coordinate of the cart with respect to the center of the rack.

- **Continuous mountain car:** In the mountain car environment, the basic task is to climb the mountain with the least energy consumption. During policy customization, we enforce an additional preference to avoid negative actions whenever possible. This corresponds to use cases where certain actions are less favorable. To monitor the change in action distribution, we count the number of negative actions executed during each episode, denoted as $n_{\text{neg}}$.

- **Highway navigation:** In the highway environment, the basic task is to navigate the vehicle safely and efficiently through the traffic. The vehicle can switch to arbitrary lanes to overtake the other vehicles. During policy customization, we enforce an additional preference to stay on the rightmost lane whenever possible. This reproduces driving habits that human drivers may have in real life. For example, drivers may prefer to drive on the rightmost lane to avoid missing the exits or when they want to drive the car slowly. We compute the average normalized lane index over an episode to validate if the desired behavior emerges, which is defined as $\bar{I}_{\text{lane}} = \sum_{t=1}^{T} I_{\text{lane},t}/2T$, where $I_{\text{lane},t} \in \{0, 1, 2\}$ is the lane index with the leftmost lane as the lane 0 and the rightmost lane as the lane 2.

- **Parking:** In the parking environment, the basic task is to park the car in the target parking slot. During policy customization, we add an additional requirement to avoid touching the boundaries of the parking slots during parking, which is considered good practice to avoid scratching other cars and to leave sufficient space for neighboring parking slots. We compute the non-violation rate to monitor if the requirement is met, which is defined as the percentage of episodes where the constraint is not violated, denoted as $\gamma_{\text{no-violation}}$.

**Baselines.** In our experiments, we mainly compare the performance of four categories of policies:

- **RL prior policy**: We train an RL policy optimizing the basic reward. It serves as the prior policy for policy customization. For environments with discrete action space (i.e., *CartPole* and *Highway*), we train the RL prior policy using soft Q-learning. For environments with continuous action space (i.e., *Continuous Mountain Car* and *Parking*), soft actor-critic is used. At test time, we also evaluate its performance on the overall task without customization, which serves as a baseline to show the effectiveness of the proposed algorithms.

- **IL prior policy**: We train another IL prior policy that imitates the RL prior with GAIL [22]. As discussed in Sec. 2, the proposed RQL algorithms are primarily motivated to customize imitative policy without knowledge of its inherent reward. The IL prior policy serves as a baseline for comparison with the residual-Q policy customized from the IL prior policy. In GAIL, we use soft Q-learning as the policy learning algorithm when the action space is discrete, and use soft actor-critic when the action space is continuous.

- **Residual-Q customized policies (Ours)**: In each environment, we train two residual-Q customized policies leveraging the RL and IL prior policies respectively. They are mainly compared against the corresponding prior policies to validate the effectiveness of policy customization. We apply residual soft Q-learning for environments with discrete action space and apply the residual soft actor-critic for environments with continuous action space. Besides, we evaluate the zero-shot customization ability of the proposed residual maximum-entropy MCTS algorithm in the *Highway* environment.

- **RL with total reward**: We also train an RL policy with the total reward (i.e., $r + \omega r_R$). Note that we do not intend to treat it as a baseline for fair comparison under the policy customization setting. We aim to compare the residual-Q policies with it to validate that the customized policies indeed solve the overall task $\hat{\mathcal{M}}$.

**Metric.** We are mainly interested in validating the performance of the proposed policy customization methods from two perspectives: 1) whether the policy is customized toward the add-on reward; 2) whether the customized policy still possesses the characteristics of the prior policy, in other words, maintains the same level of performance on the basic task. Hence, we use the average basic reward and the success rate of completing the basic task as metrics to evaluate the policies' performance on

the basic task. Meanwhile, we use the average add-on reward and task-specific metrics introduced above as the metrics for evaluating the policies' performance with respect to the customized objective.

## 5.2 RL Offline Customization

The experimental results of RL offline customization are summarized in Table 1. In all the environments and with either RL or IL prior policies, the customized policies significantly outperform the prior policies on the add-on task objectives. Meanwhile, the customized policies maintain the same level of performance as the prior policies on the basic tasks. It validates that the proposed residual-Q algorithms can tailor the policies for customized requirements while maintaining their original characteristics. One interesting observation is that the policy customization improves the performance of the IL policy on the basic task in the *Parking* environment. One factor is that GAIL does not perform well in the first place. The *Parking* environment is indeed more challenging than the other environments given the continuous action space and, more importantly, the nature of the task—navigation in tight space under the non-holonomic constraint. Another aspect we think can explain this phenomenon is that the add-on reward *implicitly* guides the policy to improve its basic task performance. As shown in Appendix G, it is difficult for the imitative policy to stop the vehicle accurately at the target parking slot. Since GAIL merely learns to match the trajectory distributions under the expert and learned policies without the knowledge of the actual task objective, it is challenging for the imitative policy to precisely localize such a narrow parking space. In contrast, the add-on boundary violation constraint confines the vehicle trajectory within the parking space once the vehicle approaches the target parking slot. Hence, the policy is able to reach the goal point faster and receives higher returns even on the basic parking task.

In Table 1, we also compare the customized policies against the RL policies trained with the total reward function, named *RL Full Policy* in the table. In environments with discrete action spaces, the residual soft-Q learning policies behave very similarly to the RL full policies. Meanwhile, the residual soft actor-critic agents have different reward patterns compared to the RL full policies. We think the behavior difference is mainly due to the approximation errors in value and policy networks. The proposed residual-Q framework relies on the prior policy to acquire knowledge of the basic task. Thus, a suboptimal prior will affect the performance of the customized policy, especially on the basic task. It also explains the performance difference between the policies customized from RL and IL priors. Inspired by the observation mentioned above on IL policy customization, we think a practical solution to mitigate this issue is partially incorporating some basic task objectives into the add-on reward if available. It is reasonable for many practical use cases even with an imitative prior. While it is difficult to precisely describe the underlying reward function of human experts, we can comfortably assume that they follow some obvious commonsense objectives (e.g., collision avoidance). In this way, our method can be considered a more principled framework for the general synergy between IL and RL beyond policy customization where the reward functions are completely decomposed.

## 5.3 MCTS Online Customization

We choose the *Highway* environment to demonstrate the zero-shot online customization ability of residual maximum-entropy MCTS. We implement residual maximum-entropy MCTS based on the MCTS planner provided by the *highway-env* environment. The planner leverages the ground-truth dynamics model of the environment for roll-out simulation. The results are summarized in Table 1. MCTS online customization is able to achieve similar performance as the RL offline counterpart.

## 5.4 Ablation Study: Comparison with RL Fine-tuning

In our RQL framework, we formulate policy customization as an MDP whose reward function balances the utilities of the imitative and add-on tasks. It provides the theoretical foundation to determine the synergy between the IL and RL objectives in a principled manner. As discussed in Sec. 1 and Sec. 4, different approaches have been proposed in the RL fine-tuning literature to combine IL and RL objectives. We investigate two representative methods under the context of policy customization. We compare them against RQL and summarize the experimental results in this subsection. Please refer to Appendix F for the complete ablation study.

**Greedy Reward Decomposition.** One common approach is regularizing the divergence between the trained RL and prior policies during policy updates. It has been widely used in offline RL algorithms

to address the distributional shift issue [30, 54], and Advantage Weighted Actor-Critic (AWAC) [40] demonstrates the benefits of such policy regularization in stabilizing and accelerating RL fine-tuning. As shown in AWAC [40], when using KL-divergence to measure the distance between policies, the regularized optimal policy becomes a maximum-entropy policy with the advantage weighted by the prior policy. If we directly adapt it to solve the policy customization problem, the customized policy is essentially defined as the solution to the following optimization problem at each policy update step:

$$\tilde{\pi}_t = \arg\max_{\tilde{\pi} \in \Pi} \mathbb{E}_{\boldsymbol{a} \sim \tilde{\pi}(\cdot|\boldsymbol{s})} \left[ \tilde{Q}_{R,t}(\boldsymbol{s}, \boldsymbol{a}) - \hat{\alpha} \log \tilde{\pi}(\boldsymbol{a}|\boldsymbol{s}) \right] - \lambda \mathcal{D}_{\mathrm{KL}}(\tilde{\pi}(\cdot|\boldsymbol{s}) \| \pi(\cdot|\boldsymbol{s})), \qquad (15)$$

where $\tilde{Q}_{R,t}$ is the soft Q-function of the MDP with only the add-on reward $r_R(\boldsymbol{s}, \boldsymbol{a})$, which is defined iteratively with the following update rule:

$$\tilde{Q}_{R,t+1}(\boldsymbol{s}, \boldsymbol{a}) = r_R(\boldsymbol{s}, \boldsymbol{a}) + \gamma \mathbb{E}_{\boldsymbol{s}' \sim p(\boldsymbol{s}'|\boldsymbol{a})} \left[ \mathbb{E}_{\boldsymbol{a}' \sim \hat{\pi}_t} \left[ \tilde{Q}_{R,t}(\boldsymbol{s}', \boldsymbol{a}') - \hat{\alpha} \log \tilde{\pi}_t(\boldsymbol{a}'|\boldsymbol{s}') \right] \right]. \qquad (16)$$

The closed-form solution to the optimization problem is given as:

$$\tilde{\pi}_t(\boldsymbol{a}|\boldsymbol{s}) \propto \exp \left( \frac{1}{\hat{\alpha} + \lambda} \left( \tilde{Q}_{R,t}(\boldsymbol{s}, \boldsymbol{a}) + \lambda \log \pi(\boldsymbol{a}|\boldsymbol{s}) \right) \right). \qquad (17)$$

We can reduce our RQL policy to the same form by simply replacing the residual Q-function in Eqn. (6) with the soft Q-function $\tilde{Q}_{R,t}$. The resulting policy solves the optimization problem in Eqn. (15) but with an entropy weight of $\hat{\alpha} - \lambda$ in the objective function. If we consider it for the policy customization problem, it is essentially equivalent to heuristically estimating the optimal Q-function as the sum of the optimal Q-functions corresponding to the basic and add-on rewards. Hence, the synthesized policy is not the optimal solution to the target MDP but a greedy approximation [52, 26]. We compare RQL with this greedily customized policy on the *Continuous Mountain Car* and *Parking* environments. The results can be found in Appendix F.1. We found that this greedy customized policy performed much worse than RQL in the most challenging *Parking* environment. Compared to the prior policy, greedy customization compromises the policy's success rate on the basic parking task to customize them towards satisfying the add-on objective. However, the greedy policy still has a significantly lower non-violation rate compared to the RQL policy. It validates that our framework is a more principled method to jointly optimize the IL and RL objectives for policy customization than divergence-based regularization.

**Augmenting Reward with Policy KL-Divergence.** Alternatively, some works [54, 61] have explored adding the policy divergence to the reward function. It has been shown to outperform policy regularization in offline RL setting [54]. To adopt this line of approaches for policy customization, we formulate the reward function of the target MDP as the sum of the add-on reward and a policy divergence regularization term. In particular, we follow [61] to augment the add-on reward with a penalty whose expectation regularizes the KL-divergence between the customized and prior policies. Formally, the reward function is defined as:

$$\check{r}_t(\boldsymbol{s}, \boldsymbol{a}) = r_R(\boldsymbol{s}, \boldsymbol{a}) - \beta \log \frac{\check{\pi}_t(\boldsymbol{a}|\boldsymbol{s})}{\pi(\boldsymbol{a}|\boldsymbol{s})}, \qquad (18)$$

where $\check{\pi}_t$ denotes the customized policy after the $t^{\mathrm{th}}$ iteration of policy update, and $\beta$ is a hyperparameter to balance the add-on reward and the policy regularization term. We can then apply standard RL algorithms to customize the policy toward this reward function. However, we found that it made the learning task much more difficult in our policy customization problem—as shown in Appendix F.2, soft actor-critic, for example, failed to find a customized policy that at least performed well on the basic parking task. We believe that one crucial factor contributing to the failure of divergence-based regularization is that the basic and add-on rewards are mostly *orthogonal* in our setting. The add-on reward only encodes additional task requirements; thus, we heavily rely on the prior policy to embed the desirable behavior on the basic task. Merely regularizing the KL-divergence is therefore insufficient—being close to the prior policy in terms of probabilistic distance does not necessarily imply behavior that accomplishes the basic task objective. Besides, incorporating the policy likelihood into the reward function makes it more challenging to update the Q-function in practice. For example, in Appendix F.3, we show that our RQL algorithm is theoretically equivalent to directly solving an MDP whose reward function is a weighted sum of the add-on reward and the log-likelihood of the prior policy. However, similar to the case of divergence-augmented reward, soft actor-critic failed to find a solution in *Parking* environment.

Table 1: Experimental Results of Residual-Q Policy Customization

| Env. | Policy | Basic Task | | Add-on Task | |
|---|---|---|---|---|---|
| | | Succ. Rate | Basic Reward | $\bar{e}_x$ | Add-on Reward |
| CartPole | RL Prior Policy | 100% | $427.21 \pm 2.25$ | $0.15 \pm 0.11$ | $-31.11 \pm 22.82$ |
| | RL Customized | 100% | $425.73 \pm 2.74$ | $\mathbf{0.05 \pm 0.05}$ | $\mathbf{-11.18 \pm 10.12}$ |
| | IL Prior Policy | 100% | $422.17 \pm 4.19$ | $0.19 \pm 0.12$ | $-38.57 \pm 24.26$ |
| | IL Customized | 100% | $423.37 \pm 4.28$ | $\mathbf{0.07 \pm 0.06}$ | $\mathbf{-14.12 \pm 12.09}$ |
| | RL Full Policy | 100% | $425.27 \pm 3.10$ | $0.05 \pm 0.04$ | $-10.72 \pm 7.43$ |
| Env. | Policy | Succ. Rate | Basic Reward | $n_{\mathrm{neg}}$ | Add-on Reward |
| Cont. Mt. Car | RL Prior Policy | 100% | $95.78 \pm 0.64$ | $42.26 \pm 1.29$ | $-4.23 \pm 0.13$ |
| | RL Customized | 100% | $95.61 \pm 0.43$ | $\mathbf{37.90 \pm 0.75}$ | $\mathbf{-3.79 \pm 0.07}$ |
| | IL Prior Policy | 100% | $92.84 \pm 2.35$ | $64.40 \pm 21.03$ | $-6.44 \pm 2.10$ |
| | IL Customized | 100% | $94.41 \pm 0.06$ | $\mathbf{41.08 \pm 1.01}$ | $\mathbf{-4.11 \pm 0.10}$ |
| | RL Full Policy | 100% | $95.14 \pm 0.21$ | $27.86 \pm 3.87$ | $-2.79 \pm 0.39$ |
| Env. | Policy | Succ. Rate | Basic Reward | $\bar{I}_{\mathrm{lane}}$ | Add-on Reward |
| Highway | RL Prior Policy | 97.22% | $44.67 \pm 2.43$ | $0.46 \pm 0.28$ | $9.17 \pm 5.54$ |
| | RL Customized | 97.19% | $42.23 \pm 1.55$ | $\mathbf{0.97 \pm 0.05}$ | $\mathbf{19.42 \pm 0.99}$ |
| | RL ME-MCTS | 98.00% | $42.71 \pm 1.65$ | $\mathbf{0.96 \pm 0.07}$ | $\mathbf{19.19 \pm 1.38}$ |
| | IL Prior Policy | 96.04% | $40.03 \pm 0.02$ | $0.50 \pm 0.40$ | $10.06 \pm 8.07$ |
| | IL Customized | 93.00% | $40.36 \pm 0.34$ | $\mathbf{0.99 \pm 0.01}$ | $\mathbf{19.88 \pm 0.24}$ |
| | IL ME-MCTS | 97.00% | $40.68 \pm 0.46$ | $\mathbf{0.99 \pm 0.01}$ | $\mathbf{19.85 \pm 0.30}$ |
| | RL Full Policy | 96.35% | $41.98 \pm 1.53$ | $0.97 \pm 0.01$ | $19.49 \pm 1.24$ |
| Env | Policy | Succ. Rate | Basic Reward | $\gamma_{\mathrm{no\text{-}violation}}$ | Add-on Reward |
| Parking | RL Prior Policy | 99.09% | $-7.08 \pm 2.65$ | 57.61% | $-1.55 \pm 3.71$ |
| | RL Customized | 98.73% | $-7.60 \pm 3.07$ | $\mathbf{96.09\%}$ | $\mathbf{-0.03 \pm 0.20}$ |
| | IL Prior Policy | 70.91% | $-12.37 \pm 9.05$ | 11.27% | $-4.94 \pm 5.52$ |
| | IL Customized | $\mathbf{83.59\%}$ | $\mathbf{-7.93 \pm 4.84}$ | $\mathbf{66.07\%}$ | $\mathbf{-0.32 \pm 0.73}$ |
| | RL Full Policy | 99.34% | $-6.81 \pm 2.39$ | 74.24% | $-0.38 \pm 0.72$ |

The evaluation results are computed over 4000 episodes for model-free policies and 100 episodes for maximum-entropy MCTS (ME-MCTS). The statistics with $\pm$ are in the format of mean $\pm$ std.

## 6 Limitations

As discussed in Sec. 5.2, one limitation of our current algorithms is that the performance of policy customization is *bottlenecked* by the imitative prior policy. Policy customization not only requires it to perform well on the trajectory distribution under the imitative prior policy itself but, more importantly, we need the imitative policy to accurately indicate desired behavior with respect to the basic task objective on the trajectory distribution under the *customized* policy. Hence, without careful design, our algorithms would suffer from the distributional shift issue. While, in practice, this issue can be mitigated by partially incorporating some basic task objectives into the add-on reward if available, a more principled and fundamental solution is to obtain a diverse imitative prior that is best suited for customization. Generative models, such as normalizing flows [48] and, more recently, diffusion models [23, 6, 20], have been shown promising in constructing such a diverse behavior prior. Besides, incorporating human experts in the loop has also been shown to be effective in improving the generalizability of imitation learning [27, 38, 41, 33]. We will investigate their applications in the policy customization problem in future work. Also, we only demonstrate MCTS online customization with an ideal perfect dynamics model in our current experiments. In future work, we will combine residual maximum-entropy MCTS with offline learned predictive models to achieve zero-shot online customization in more practical scenarios.

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
