# A  Residual Q-Function Derivation

In this section, we provide a detailed breakdown of the derivation process of the residual Q-function, i.e., Eqn. (5). First, we can write $r(\boldsymbol{s}, \boldsymbol{a})$ as a function of the soft Q-function according to Eqn. (2):

$$r(\boldsymbol{s}, \boldsymbol{a}) = \gamma \mathbb{E}_{\boldsymbol{s}' \sim p(\cdot|\boldsymbol{s}, \boldsymbol{a})} \left[ \alpha \log \int_{\mathcal{A}} \exp\left(\frac{1}{\alpha} Q^*(\boldsymbol{s}', \boldsymbol{a}')\right) d\boldsymbol{a}' \right] - Q^*(\boldsymbol{s}, \boldsymbol{a}). \tag{19}$$

Eqn. (5b) results from substituting $r(\boldsymbol{s}, \boldsymbol{a})$ with the right-hand side of the above equation. To derive Eqn. (5c), we first write $\int_{\mathcal{A}} \exp\left(\frac{1}{\alpha} Q^*(\boldsymbol{s}', \boldsymbol{a}')\right) da$ as the normalization factor $Z_{s'}$ based on its definition. Then note that, according to the definition of the residual Q-function $Q_{R,t}$, we can substitute $\hat{Q}_t(\boldsymbol{s}', \boldsymbol{a}')$ in Eqn. (5b) with:

$$\begin{aligned}
\hat{Q}_t(\boldsymbol{s}', \boldsymbol{a}') &= Q_{R,t}(\boldsymbol{s}', \boldsymbol{a}') + \omega Q^*(\boldsymbol{s}', \boldsymbol{a}') \\
&= Q_{R,t}(\boldsymbol{s}', \boldsymbol{a}') + \omega\alpha \log \pi\left(\boldsymbol{a}'|\boldsymbol{s}'\right) + \omega\alpha \log Z_{s'},
\end{aligned} \tag{20}$$

where the second step comes from Eqn. (3). To obtain Eqn. (5d), note that $Z_{s'}$ is not a function of $a'$, so we can write the last term of Eqn. (5c) as:

$$\mathbb{E}_{s'} \left[ \hat{\alpha} \log \int_{\mathcal{A}} \exp\left(\frac{1}{\hat{\alpha}} \left(Q_{R,t}(\boldsymbol{s}', \boldsymbol{a}') + \omega\alpha \log \pi\left(\boldsymbol{a}'|\boldsymbol{s}'\right) + \omega\alpha \log Z_{s'}\right)\right) d\boldsymbol{a}' \right]$$

$$= \mathbb{E}_{s'} \left[ \hat{\alpha} \log Z_{s'}^{\omega\alpha/\hat{\alpha}} \int_{\mathcal{A}} \exp\left(\frac{1}{\hat{\alpha}} \left(Q_{R,t}(\boldsymbol{s}', \boldsymbol{a}') + \omega\alpha \log \pi\left(\boldsymbol{a}'|\boldsymbol{s}'\right)\right)\right) d\boldsymbol{a}' \right] \tag{21a}$$

$$= \mathbb{E}_{s'} \left[ \hat{\alpha} \log \int_{\mathcal{A}} \exp\left(\frac{1}{\hat{\alpha}} \left(Q_{R,t}(\boldsymbol{s}', \boldsymbol{a}') + \omega\alpha \log \pi\left(\boldsymbol{a}'|\boldsymbol{s}'\right)\right)\right) d\boldsymbol{a}' \right] + \omega\alpha\gamma \mathbb{E}_{s'} \log Z_{s'}. \tag{21b}$$

# B  Residual Soft Q-learning Derivation

In this section, we provide the detailed derivation of residual soft Q-learning's objective function, i.e., Eqn (7). The original objective of soft Q-learning is to minimize the TD error of the soft Q-function:

$$J_Q(\theta) = \mathbb{E}_{(\boldsymbol{s}_t, \boldsymbol{a}_t)} \left[ \frac{1}{2} \left( \hat{Q}_{\bar{\theta}}^{\text{target}}(\boldsymbol{s}_t, \boldsymbol{a}_t) - \hat{Q}_\theta(\boldsymbol{s}_t, \boldsymbol{a}_t) \right)^2 \right], \tag{22}$$

where the target Q-value is defined as:

$$\hat{Q}_{\bar{\theta}}^{\text{target}}(\boldsymbol{s}_t, \boldsymbol{a}_t) = r_R(\boldsymbol{s}_t, \boldsymbol{a}_t) + \omega r(\boldsymbol{s}_t, \boldsymbol{a}_t) + \gamma \mathbb{E}_{\boldsymbol{s}' \sim p(\cdot|\boldsymbol{s}_t, \boldsymbol{a}_t)} \left[ \hat{\alpha} \log \int_{\mathcal{A}} \exp\left(\frac{1}{\hat{\alpha}} \hat{Q}_\theta(\boldsymbol{s}', \boldsymbol{a}')\right) d\boldsymbol{a}' \right]. \tag{23}$$

Given a parameterized residual Q-function, we can write the parameterized soft Q-function as $\hat{Q}_\theta(\boldsymbol{s}, \boldsymbol{a}) = Q_{R,\theta}(\boldsymbol{s}, \boldsymbol{a}) + \omega Q^*(\boldsymbol{s}, \boldsymbol{a})$. Following steps similar to Eqn. (5), we can express the target Q-value with the residual Q-function as:

$$\begin{aligned}
\hat{Q}_{\bar{\theta}}^{\text{target}}(\boldsymbol{s}_t, \boldsymbol{a}_t) =& r_R(\boldsymbol{s}_t, \boldsymbol{a}_t) + \omega Q^*(\boldsymbol{s}_t, \boldsymbol{a}_t) \\
&+ \gamma \mathbb{E}_{\boldsymbol{s}' \sim p(\cdot|\boldsymbol{s}_t, \boldsymbol{a}_t)} \left[ \hat{\alpha} \log \int_{\mathcal{A}} \exp\left(\frac{1}{\hat{\alpha}} \left(Q_{R,\theta}(\boldsymbol{s}', \boldsymbol{a}') + \omega' \log \pi(\boldsymbol{a}'|\boldsymbol{s}')\right)\right) d\boldsymbol{a}' \right].
\end{aligned} \tag{24}$$

Thus, if we define the target residual Q-value as:

$$\begin{aligned}
\hat{Q}_{R,\bar{\theta}}^{\text{target}}(\boldsymbol{s}_t, \boldsymbol{a}_t) =& r_R(\boldsymbol{s}_t, \boldsymbol{a}_t) \\
&+ \gamma \mathbb{E}_{\boldsymbol{s}' \sim p(\cdot|\boldsymbol{s}_t, \boldsymbol{a}_t)} \left[ \hat{\alpha} \log \int_{\mathcal{A}} \exp\left(\frac{1}{\hat{\alpha}} \left(Q_{R,\theta}(\boldsymbol{s}', \boldsymbol{a}') + \omega' \log \pi(\boldsymbol{a}'|\boldsymbol{s}')\right)\right) d\boldsymbol{a}' \right],
\end{aligned} \tag{25}$$

it is then straightforward to see that the TD error of the soft Q-function equals the TD error of the residual Q-function. It leads to Eqn. (7) and (8) when we use the data in the replay buffer to estimate the target residual Q-value and the TD error.

## C   Policy Evaluation for Residual Soft Actor-Critic

In this section, we derive the policy evaluation step of the residual soft actor-critic algorithm introduced in Sec. 3. In the soft actor-critic algorithm, the policy evaluation step aims to iteratively compute the soft Q-value of a policy $\hat{\pi}$, which relies on repeatedly applying a modified soft Bellman backup operator given by:

$$\hat{Q}_{t+1}(\boldsymbol{s}, \boldsymbol{a}) = r_R(\boldsymbol{s}, \boldsymbol{a}) + \omega r(\boldsymbol{s}, \boldsymbol{a}) + \gamma \mathbb{E}_{\boldsymbol{s}' \sim p(\boldsymbol{s}'|\boldsymbol{a})} \left[ \mathbb{E}_{\boldsymbol{a}' \sim \hat{\pi}} \left[ \hat{Q}_t(\boldsymbol{s}', \boldsymbol{a}') - \hat{\alpha} \log \hat{\pi}(\boldsymbol{a}'|\boldsymbol{s}') \right] \right]. \quad (26)$$

We can then derive an update rule for the residual Q-function from the modified soft Bellman backup operator following a similar procedure as in Eqn. (5):

$$Q_{R,t+1}(\boldsymbol{s}, \boldsymbol{a})$$
$$= r_R(\boldsymbol{s}, \boldsymbol{a}) + \omega r(\boldsymbol{s}, \boldsymbol{a}) + \gamma \mathbb{E}_{\boldsymbol{s}'} \left[ \mathbb{E}_{\boldsymbol{a}' \sim \hat{\pi}} \left[ \hat{Q}_t(\boldsymbol{s}', \boldsymbol{a}') - \hat{\alpha} \log \hat{\pi}(\boldsymbol{a}'|\boldsymbol{s}') \right] \right] - \omega Q^*(\boldsymbol{s}, \boldsymbol{a}), \quad (27a)$$
$$= r_R(\boldsymbol{s}, \boldsymbol{a}) + \omega Q^*(\boldsymbol{s}, \boldsymbol{a}) - \omega' \gamma \mathbb{E}_{\boldsymbol{s}'} \log Z_{s'} - \omega Q^*(\boldsymbol{s}, \boldsymbol{a})$$
$$\quad + \gamma \mathbb{E}_{\boldsymbol{s}'} \left[ \mathbb{E}_{\boldsymbol{a}' \sim \hat{\pi}} \left[ Q_{R,t}(\boldsymbol{s}', \boldsymbol{a}') + \omega' \log \pi(\boldsymbol{a}'|\boldsymbol{s}') + \omega' \log Z_{s'} - \hat{\alpha} \log \hat{\pi}(\boldsymbol{a}'|\boldsymbol{s}') \right] \right], \quad (27b)$$
$$= r_R(\boldsymbol{s}, \boldsymbol{a}) - \omega' \gamma \mathbb{E}_{\boldsymbol{s}'} \log Z_{s'} + \omega' \gamma \mathbb{E}_{\boldsymbol{s}'} \log Z_{s'}$$
$$\quad + \gamma \mathbb{E}_{\boldsymbol{s}'} \left[ \mathbb{E}_{\boldsymbol{a}' \sim \hat{\pi}} \left[ Q_{R,t}(\boldsymbol{s}', \boldsymbol{a}') + \omega' \log \pi(\boldsymbol{a}'|\boldsymbol{s}') - \hat{\alpha} \log \hat{\pi}(\boldsymbol{a}'|\boldsymbol{s}') \right] \right], \quad (27c)$$
$$= r_R(\boldsymbol{s}, \boldsymbol{a}) + \gamma \mathbb{E}_{\boldsymbol{s}'} \left[ \mathbb{E}_{\boldsymbol{a}' \sim \hat{\pi}} \left[ Q_{R,t}(\boldsymbol{s}', \boldsymbol{a}') + \omega' \log \pi(\boldsymbol{a}'|\boldsymbol{s}') - \hat{\alpha} \log \hat{\pi}(\boldsymbol{a}'|\boldsymbol{s}') \right] \right]. \quad (27d)$$

## D   Environment Configuration

In this section, we introduce the detailed configurations of the selected environments, including the definitions of the state and action spaces, observations, and reward functions.

**CartPole balancing**. In the cartpole environment, a pole is attached to a cart by an unactuated joint. The cart is moving along a track. The states are defined as the coordinate of the cart along the track, $x$, the angle of the pole, $\delta$, the velocity of the cart, $v$, and the angular velocity of the pole, $\dot{\delta}$. The action is a binary variable indicating the direction of the force exerted on the cart, i.e., $\mathcal{A} = \{\text{"left"}, \text{"right"}\}$. The goal of the basic task is to balance the pole by exerting forces on the cart, which is considered to be successful if the pole does not fall down for 500 steps. The basic reward function is a sum of two components, which are:

$$\text{Survival Reward}: \quad r_{\text{survival}}(\boldsymbol{s}, \boldsymbol{a}) = 1, \quad (28)$$

$$\text{Balancing Reward}: \quad r_{\text{balance}}(\boldsymbol{s}, \boldsymbol{a}) = -\frac{10|\delta|}{0.2095}. \quad (29)$$

During policy customization, we demand an additional task that requires the cart to stay at the center of the rack. The corresponding add-on reward is defined as $r_R(\boldsymbol{s}, \boldsymbol{a}) = -|x|/2.4$, which penalizes the deviation from the center of the rack at each time step.

**Continuous mountain car**. In the mountain car environment, the car is placed at the bottom of a sinusoidal valley with a randomized initial position. The states are defined as the horizontal coordinate of the car, $x$, and the velocity of the car, $v$. The action is the directional force $f$ applied to the car. The goal of the basic task is to accelerate the car to reach the goal state on top of the right hill with the least energy consumption. It is considered successful if the car reaches the goal within 999 steps. The basic reward function is designed as a sum of the following two components:

$$\text{Goal Reward}: \quad r_{\text{goal}}(\boldsymbol{s}, \boldsymbol{a}) = 100 \times \mathbb{1}(\boldsymbol{s} = \boldsymbol{s_g}), \quad (30)$$
$$\text{Energy Cost}: \quad r_{\text{energy}}(\boldsymbol{s}, \boldsymbol{a}) = -0.1 f^2. \quad (31)$$

During policy customization, we enforce an additional preference to avoid negative actions whenever possible. The add-on reward is defined as $r_R(\boldsymbol{s}, \boldsymbol{a}) = -0.5 \times \mathbb{1}(f < 0)$ to penalize negative force.

**Highway navigation**. In the highway environment, we navigate the vehicle on a three-lane highway around other vehicles. The state vector of the $i$-th vehicle is defined as $\boldsymbol{s}_i = \left[ x_i, y_i, v_{x,i}, v_{y,i}, \theta_i, \dot{\theta}_i \right]^\mathsf{T}$, where $x_i, y_i$ are the xy-coordinates, $v_{x,i}, v_{y,i}$ are the velocities along the x- and y-axes, $\theta_i$ is the yaw

angle, and $\dot{\theta}$ is the yaw rate. The ego vehicle controlled by the policy is indexed with $i = 0$. The observation collects the normalized states of all the vehicles. The discrete action space consists of five meta-actions that govern the corresponding built-in controllers provided as part of the environment. The meta-actions define different high-level behaviors in the highway environment, referred to as "switch left", "switch right", "faster", "idle", and "slower" [30]. The goal of the basic task is to drive the vehicle through the traffic safely and efficiently. The task is considered successful if the car is driven without collision over 40 steps. The basic reward function consists of three components:

$$\text{Survival Reward}: \quad r_{\text{survival}}(\boldsymbol{s}, \boldsymbol{a}) = 1, \tag{32}$$

$$\text{Velocity Reward}: \quad r_{\text{velocity}}(\boldsymbol{s}, \boldsymbol{a}) = 0.4 v_{\text{norm}}, \tag{33}$$

$$\text{Collision Cost}: \quad r_{\text{collision}}(\boldsymbol{s}, \boldsymbol{a}) = -0.5 \times \mathbb{1}(\text{isCollision}(\boldsymbol{s}) = 1), \tag{34}$$

where $v_{\text{norm}}$ is defined as $(\sqrt{v_{x,0}^2 + v_{y,0}^2} - 20)/10$ and clipped into $[0, 1]$, and isCollision is a built-in function to determine if the ego collides with the other vehicles. During policy customization, we enforce an additional preference to stay on the rightmost lane whenever possible, which is formulated as an add-on reward function defined as:

$$r_R(\boldsymbol{s}, \boldsymbol{a}) = 0.5 \frac{I_{\text{lane}}(\boldsymbol{s})}{\text{num of lanes} - 1}, \tag{35}$$

where the function $I_{\text{lane}}$ gives the index of the lane where the car is driving, with the leftmost lane indexed with 0.

**Parking**. In the parking environment, we control a vehicle in the parking lot. The states of the vehicle are defined the same as in the highway environment. The observation consists of the current state, $\boldsymbol{s}$, and the target parking state, $\boldsymbol{s}_g$. The actions are the longitudinal acceleration command, $a$, and the steering angle command, $\delta$. The goal of the basic task is to park the vehicle at the target parking space within a minimal number of time steps, the task is considered successful if the error between the vehicle state and the target parking state becomes smaller than a threshold value within 100 steps. Hence, the basic reward is simply defined as $r(\boldsymbol{s}, \boldsymbol{a}) = -\boldsymbol{w}^T \|\boldsymbol{s} - \boldsymbol{s}_g\|$. During policy customization, we add an additional requirement to avoid touching the boundaries of the parking slots during parking, the add-on reward is defined as $r_R(\boldsymbol{s}, \boldsymbol{a}) = -\mathbb{1}(\text{Violation}(\boldsymbol{s}) = 1)$, where Violation is a function detecting whether the vehicle violates the enforced boundary constraint.

# E  Implementation Details and Hyperparameters

In this section, we introduce the detailed implementation of the proposed algorithms and the hyperparameters we used in our experiments for each environment. We implemented the proposed algorithms and all the baseline methods upon Stable-Baselines3 [41] and its imitation library [16]. In addition, we implemented the residual maximum-entropy MCTS based on the MCTS planner provided by the *highway-env* environment. All the experiments were conducted on Ubuntu 16.04 with Intel Core i7-7700 CPU @ 3.60GHz × 8, GeForce GTX 1070/PCIe/SSE2, and 32 GB RAM.

## E.1  Algorithm Implementation

**Residual Soft Q-Learning** was implemented upon the standard deep Q-network (DQN) from Stable-Baselines3 [41]. In specific, we substituted the target Q-value with the target residual Q-value defined in Eqn. (8) when computing the TD error for the loss function.

**Residual Soft Actor-Critic** was implemented upon the standard soft actor-critic (SAC) from Stable-Baselines3 [41]. Similar to the case of residual soft Q-learning, we substituted the target Q-value with the target residual Q-value defined in Eqn. (9) when computing the TD error for the critic loss. In addition, we added the log-likelihood of the prior policy into the actor loss as in Eqn. (10).

**Residual Maximum-Entropy MCTS** was implemented upon the MCTS planner provided by the *highway-env* environment. We first adapted it to the maximum-entropy MCTS proposed in [52] and then implemented the residual maximum-entropy MCTS upon it. The pseudo-code of the proposed algorithm is presented in the Algorithm 1.

## E.2  Hyperparameters

**RL prior policy.** We adopted the hyperparameters from RL Baselines3 Zoo [40] for the *Cart Pole*, *Continuous Mountain Car* and *Highway* environments. For the *Parking* environment, we adopted

the hyperparameter provided in Stable-Baselines3's document [41]. We further adjusted the learning rate and increase the number of training steps for better performance. The final learning rate and the number of training steps for each environment are summarized in Table 2.

**IL prior policy.** The IL prior policies were trained by imitating the RL prior policies with GAIL. The same RL algorithms used to train the RL experts were used as the policy learning algorithms in GAIL. Thus, we adopted the same hyperparameters for the policy learners. For each environment, we further tuned the following hyperparameters specified in the Stable-Baselines3 imitation library [16]:

- `expert_min_episodes`: The minimum number of episodes of expert demonstration.

- `demo_batch_size`: The number of samples contained in each batch of expert data.

- `gen_replay_buffer_capacity`: The capacity of the generator replay buffer, i.e., the maximum number of state-action pairs sampled from the generator that can be stored.

- `n_disc_updates_per_round`: The number of discriminator update steps after each iteration of generator update.

The final values of these hyperparameters for each environment are summarized in Table 3.

**Residual Q-learning policies.** The model-free residual Q-learning policies were trained with the same hyperparameters of their corresponding RL prior policy. The additional hyperparameters are summarized in Table 4. For the *Parking* environment, we scaled $\omega'$ by 10 times at the first $10^5$ training steps, so that the residual Q-learning policy was trained to copy the prior policy. We found that it could accelerate the exploration at the early stage and stabilize the training process. For residual maximum-entropy MCTS, the hyperparameters include $\omega'$, maximum iterations $Iter_{\max}$, planning horizon $H$, and the exploration coefficient $\epsilon$. The final values of hyperparameters we used in the *Parking* experiment are summarized in Table 5.

Table 2: Learning Rate and Training Steps of RL Policies

| Hyperparameter | Cart Pole | Mountain Car Continuous | Highway | Parking |
|---|---|---|---|---|
| Learning Rate | $2.3 \times 10^{-3}$ | $3 \times 10^{-4}$ | $10^{-4}$ | $10^{-3}$ |
| Number of Training Steps | $10^5$ | $10^5$ | $5 \times 10^5$ | $8 \times 10^5$ |

Table 3: Hyperparameters of GAIL Imitated Policies

| Hyperparameter | Cart Pole | Mountain Car Continuous | Highway | Parking |
|---|---|---|---|---|
| `expert_min_episodes` | $10^3$ | $10^4$ | $10^3$ | $10^4$ |
| `demo_batch_size` | 1024 | 1024 | 1024 | 1024 |
| `gen_replay_buffer_capacity` | 2048 | 2048 | 2048 | 2048 |
| `n_disc_updates_per_round` | 4 | 4 | 4 | 4 |

Table 4: Hyperparameters of Residual Q Policies

| Hyperparameter | Prior Policy | Cart Pole | Mountain Car Continuous | Highway | Parking |
|---|---|---|---|---|---|
| $\alpha$ | RL | 1 | 0.1 | 1 | 0.0097 |
|  | IL | 1 | 0.1 | 1 | auto |
| $\hat{\alpha}$ | RL | 1 | 0.1 | 1 | auto |
|  | IL | 1 | 0.1 | 1 | 0.0611 |
| $\omega'$ | RL | 1 | 0.1 | 1 | 0.0097 |
|  | IL | 1 | 0.1 | 1 | 0.0611 |

Table 5: Hyperparameters of Residual Maximum-Entropy MCTS

| Hyperparameter | Prior Policy | Value |
|:---:|:---:|:---:|
| $\omega'$ | RL | 1 |
| | IL | 1 |
| $Iter_{\max}$ | RL | 150 |
| | IL | 150 |
| $H$ | RL | 6 |
| | IL | 6 |
| $\epsilon$ | RL | 0.1 |
| | IL | 1 |

# F    Ablation Study

In this section, we investigate three alternative algorithm designs for policy customization to illustrate the advantages of the proposed residual Q-learning framework. In Sec. F.1 and F.2, we study two alternatives where the KL-divergence between the customized and prior policies is used to incorporate the behavior prior into the customized policy. In Sec. F.3, we study a theoretically equivalent form of residual Q-learning that can be directly solved with off-the-shelf RL algorithms.

## F.1    Greedy Reward Decomposition

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

where $\check{\pi}_t$ denotes the customized policy after the $t^{\text{th}}$ iteration of policy update, and $\beta$ is a hyperparameter to balance the add-on reward and the policy regularization term. We can then apply standard RL algorithms to customize the policy toward this reward function. We evaluated this method in the *Parking* environment with soft actor-critic as the learning algorithm. As shown in Figure 2, It failed to find a customized policy that at least perform well on the basic parking task—The success rate was persistently close to zero during the learning procedure. We believe that one crucial factor contributing to its failure is that the basic and add-on rewards are mostly *orthogonal* in our setting. The add-on reward only encodes additional task requirements; thus, we heavily rely on the prior policy to embed the desirable behavior on the basic task. Merely regularizing the policy divergence is therefore insufficient—being close to the prior policy in terms of probabilistic distance does not necessarily imply behavior that accomplishes the basic task objective.

## F.3 Augmenting Reward with Policy Log Likelihood

The last alternative algorithm we investigate is a theoretically equivalent form of residual Q-learning. Observe that if we add $\omega' \log \pi(\boldsymbol{s}, \boldsymbol{a})$ to both sides of Eqn. (5), we obtain:

$$Q_{R,t+1}^{\text{aug}}(\boldsymbol{s}, \boldsymbol{a}) = r_R(\boldsymbol{s}, \boldsymbol{a}) + \omega' \log \pi(\boldsymbol{s}, \boldsymbol{a}) + \gamma \mathbb{E}_{\boldsymbol{s}'} \left[ \hat{\alpha} \log \int_{\mathcal{A}} \exp \left( \frac{1}{\hat{\alpha}} Q_{R,t}^{\text{aug}}(\boldsymbol{s}', \boldsymbol{a}') \right) d\boldsymbol{a}' \right], \quad (40)$$

where $Q_{R,t}^{\text{aug}}(\boldsymbol{s}, \boldsymbol{a}) = Q_{R,t}(\boldsymbol{s}, \boldsymbol{a}) + \omega' \log \pi(\boldsymbol{s}, \boldsymbol{a})$. It is then straightforward to see that $Q_{R,t}^{\text{aug}}$ is the soft Q-function corresponding to the reward function $r_R(\boldsymbol{s}, \boldsymbol{a}) + \omega' \log \pi(\boldsymbol{s}, \boldsymbol{a})$, which means that we can find a policy equivalent to the residual Q-learning one, simply through solving the MDP $\mathcal{M}^{\text{aug}} = (\mathcal{S}, \mathcal{A}, r_R + \omega' \log \pi, p)$ with off-the-shell RL algorithms. We test this equivalent algorithm in the *Parking* environment. As shown in Figure 2, similar to the case with divergence-augmented reward, soft actor-critic failed to find a customized policy that can complete the basic parking task. We think it is because estimating $Q_{R,t}^{\text{aug}}$ is inherently difficult since the complex policy log-likelihood function is embedded into the reward. In contrast, when estimating $Q_{R,t}^{\text{aug}}$, our residual Q-learning framework fully leverages our prior knowledge on $\log \pi$ and only estimates the residual part during policy customization. Thus, residual Q-learning can more efficiently explore and optimize the policy.

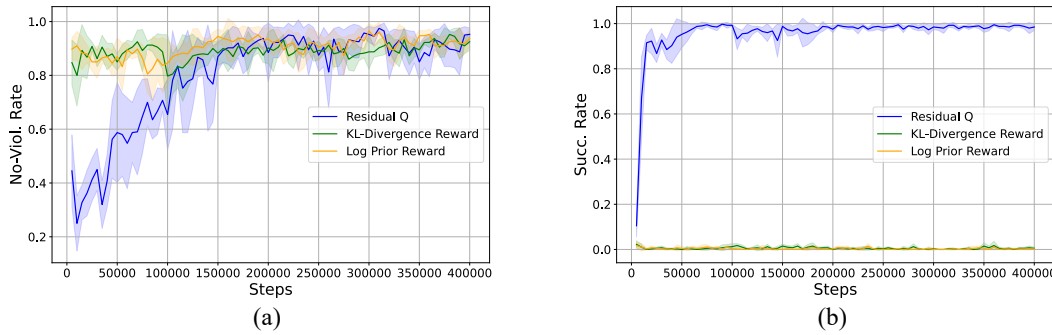

(a)                                                      (b)

Figure 2: Learning curves in the *Parking* environment for different algorithms, including residual soft actor-critic, soft actor-critic with divergence-augmented reward (Appendix F.2), and soft actor-critic with likelihood-augmented reward (Appendix F.3). We plot the curves of the non-violation rate and success rate over the number of episodic steps in subplots (a) and (b) respectively. The error bands indicate standard deviations computed over four trials with different random seeds.

Time

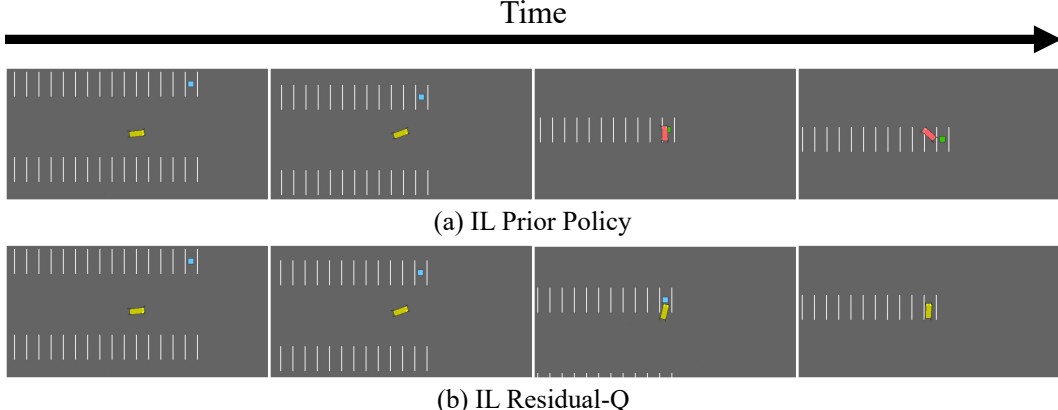

(a) IL Prior Policy

(b) IL Residual-Q

Figure 3: Representative examples from the *Parking* environment comparing the results of executing the IL prior and the residual Q-learning policy customized from the IL prior.

## G    Visualization

In this section, we visualize some representative examples from the *Parking* environment. As shown in Figure 3, the IL policy fails to stop the car within the target parking slot, while the customized policy is able to do so with the guidance of the boundary violation constraint. As shown in Figure 4, compared to the RL prior policy, the proposed residual Q-learning policy finds a very different and violation-free parking route, whereas the greedily customized policy gets stuck and fails to reach the target parking slot. Also, it is worth noting that the customized policy behaves differently from the RL full policy. As discussed in Sec. 5.2, we think the behavior difference is mainly due to the approximation errors in value and policy networks, which results in an imperfect prior policy.

## H    Additional Experiments

In this section, we present the additional experiments conducted in the MuJoCo environments [47]. The selected environments are *Ant-v3*, *Humanoid-v3*, and *Hopper-v3*. The basic task is to control the robot to move along the positive $x$-direction. During policy customization, we added an add-on reward to encourage the robot to also move along the positive $y$-direction for *Ant* and *Humanoid*. For Hopper, we added an add-on reward to encourage the robot to jump higher. The results are summarized in Table 7 and Fig. 5. In all the environments, the policies customized by residual Q-learning achieved 1) higher add-on rewards than the prior policies; and 2) a trade-off between the basic and add-on tasks similar to the RL full policies. In contrast, the RL fine-tuning baseline

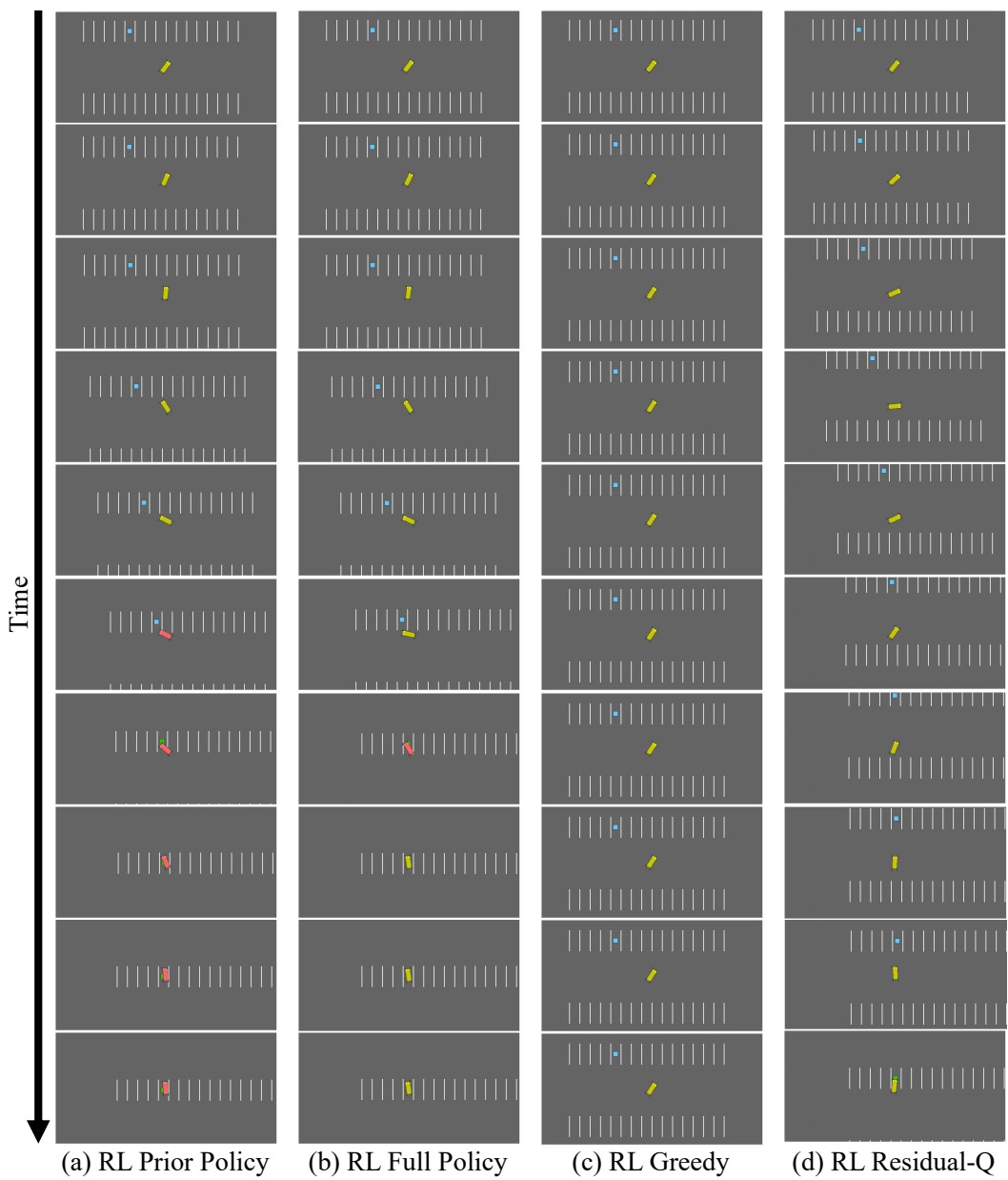

Figure 4: Representative examples from the *Parking* environment comparing the results of executing RL prior, RL full policy, and policies customized from RL prior with different approaches.

(i.e., the one described in Sec. 5.4 and Appendix F.1) tends to achieve higher add-on rewards but lower basic rewards compared to the residual-Q and RL full policies. The total rewards of the RL fine-tuning baseline are also always lower than the residual-Q policies. The results further validate that the proposed residual Q-learning method outperforms RL fine-tuning in policy customization problems. The only exception is *Humanoid* with IL prior. Note that this IL prior was trained by BC since we did not find hyperparameters to let GAIL succeed. The BC prior is not an ideal prior suitable for residual Q-learning, as it does not follow the maximum-entropy policy distribution. It is reasonable that residual-Q performs worse than RL fine-tuning given the less diverse BC prior, since residual Q-learning relies on the prior policy to encode the basic task reward, whereas RL fine-tuning only uses the prior policy as a regularization.

Table 7: Experimental Results of Residual-Q Policy Customization in Mujoco

| Env. | Policy | Full Task | Basic Task | Add-on Task | |
|---|---|---|---|---|---|
| | | Total Reward | Basic Reward | $\bar{v}_y$ | Add-on Reward |
| Ant | RL Prior Policy | $5586.71 \pm 1098.17$ | $5527.94 \pm 1075.61$ | $0.06 \pm 0.14$ | $58.77 \pm 135.27$ |
| | RL Greedy | $6373.46 \pm 689.80$ | $2528.59 \pm 350.13$ | $3.86 \pm 0.32$ | $3844.87 \pm 355.00$ |
| | RL Residual-Q | $\mathbf{6913.15 \pm 587.86}$ | $3080.87 \pm 284.93$ | $3.86 \pm 0.11$ | $3832.28 \pm 324.68$ |
| | IL Prior Policy | $5280.65 \pm 1854.30$ | $4673.23 \pm 1665.73$ | $0.64 \pm 0.22$ | $607.42 \pm 221.39$ |
| | IL Greedy | $6050.26 \pm 1403.14$ | $1962.40 \pm 531.23$ | $4.28 \pm 0.62$ | $4087.87 \pm 896.79$ |
| | IL Residual-Q | $\mathbf{6760.40 \pm 755.71}$ | $3105.70 \pm 367.35$ | $3.72 \pm 0.18$ | $3654.70 \pm 402.45$ |
| | RL Full Policy | $6642.35 \pm 1607.73$ | $3546.07 \pm 858.64$ | $3.27 \pm 0.49$ | $3096.29 \pm 761.58$ |
| Env. | Policy | Total Reward | Basic Reward | $\bar{v}_y$ | Add-on Reward |
| Humanoid | RL Prior Policy | $5514.65 \pm 59.25$ | $5472.68 \pm 32.19$ | $0.04 \pm 0.08$ | $41.97 \pm 79.92$ |
| | RL Greedy | $6209.65 \pm 1660.55$ | $4513.35 \pm 1186.40$ | $1.74 \pm 0.33$ | $1696.30 \pm 474.79$ |
| | RL Residual-Q | $\mathbf{6126.81 \pm 18.73}$ | $5363.79 \pm 17.21$ | $0.76 \pm 0.01$ | $763.02 \pm 9.79$ |
| | IL Prior Policy* | $4848.65 \pm 2278.13$ | $4874.01 \pm 2297.02$ | $-0.01 \pm 0.09$ | $-25.35 \pm 41.45$ |
| | IL Greedy | $\mathbf{9306.88 \pm 21.17}$ | $6586.47 \pm 21.01$ | $2.72 \pm 0.02$ | $2720.41 \pm 19.16$ |
| | IL Residual-Q | $7610.38 \pm 515.08$ | $5206.52 \pm 347.90$ | $2.41 \pm 0.11$ | $2403.87 \pm 167.26$ |
| | RL Full Policy | $5771.79 \pm 273.83$ | $5403.25 \pm 257.07$ | $0.37 \pm 0.01$ | $368.55 \pm 19.85$ |
| Env | Policy | Total Reward | Basic Reward | $\bar{z}$ | Add-on Reward |
| Hopper | RL Prior Policy | $4439.13 \pm 805.73$ | $3217.66 \pm 568.79$ | $1.33 \pm 0.01$ | $1221.47 \pm 237.53$ |
| | RL Greedy | $4661.77 \pm 14.30$ | $3266.62 \pm 14.39$ | $1.40 \pm 0.00$ | $1395.15 \pm 4.35$ |
| | RL Residual-Q | $\mathbf{4798.23 \pm 21.92}$ | $3428.70 \pm 18.62$ | $1.37 \pm 0.00$ | $1369.52 \pm 4.49$ |
| | IL Prior Policy | $3828.48 \pm 796.73$ | $2754.69 \pm 568.38$ | $1.32 \pm 0.01$ | $1073.79 \pm 228.62$ |
| | IL Greedy | $4619.94 \pm 13.50$ | $3236.77 \pm 13.41$ | $1.38 \pm 0.00$ | $1383.17 \pm 0.51$ |
| | IL Residual-Q | $\mathbf{4704.97 \pm 48.77}$ | $3335.57 \pm 35.53$ | $1.37 \pm 0.01$ | $1369.40 \pm 17.96$ |
| | RL Full Policy | $4698.78 \pm 7.23$ | $3242.15 \pm 5.71$ | $1.46 \pm 0.00$ | $1456.63 \pm 4.83$ |

The results are computed over 200 episodes. The statistics are in the format of mean $\pm$ std.
* The IL prior was trained by BC in *Humanoid* since we did not find hyperparameters to let GAIL succeed.

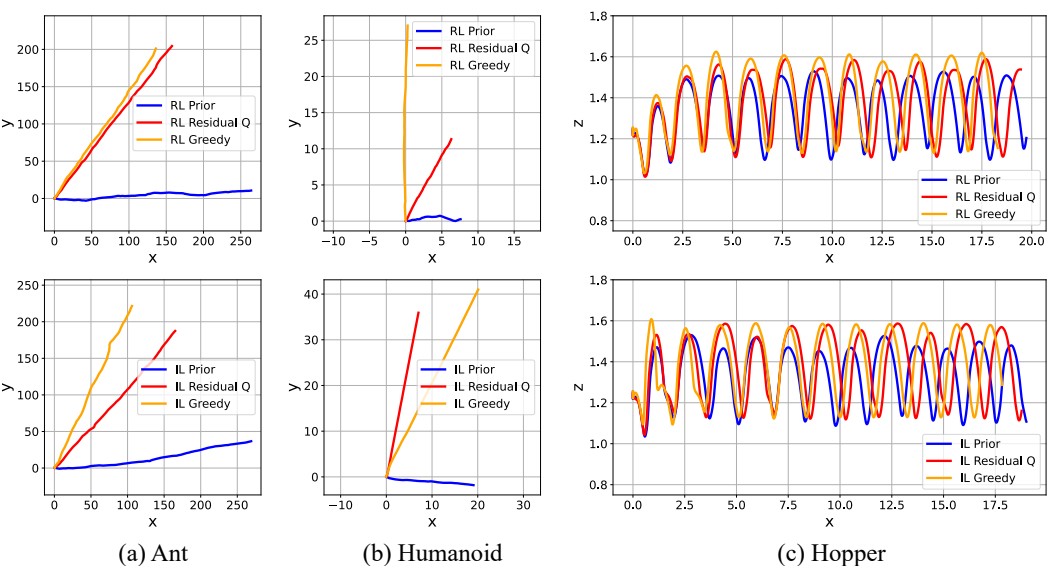

(a) Ant   (b) Humanoid   (c) Hopper

Figure 5: (a) The trajectory of the Ant robot on the $x$ and $y$ axis. (b) The trajectory of the Humanoid robot on the $x$ and $y$ axis. (c) The trajectory of the top of the Hopper robot on the $x$ and $z$ axis.

**Algorithm 1** Residual Maximum-Entropy MCTS
___
**Input:** Current state $s_0$; Prior policy $\pi$; Hyperparameter $\omega'$; Maximum iterations $Iter_{\max}$; Planning horizon $H$; Exploration coefficient $\epsilon$.
**Output:** Root node $n_0$ of the built tree.
  1: Initialize the root node $n_0(s_0)$.
  2: Initialize iteration counter $Iter = 0$.
  3: **while** $Iter < Iter_{\max}$ **do**
  4:     $n_{T-1} \leftarrow$ SELECTION$(n_0)$
  5:     $n_T \leftarrow$ EXPANSION$(n_{T-1})$
  6:     $R \leftarrow$ SIMULATION&EVALUATION$(n_T)$
  7:     BACK-PROPAGATION$(n_T, R)$
  8: **end while**
  9: **return** $n_0$

 10: **function** SELECTION$(n_0)$
 11:     $n_{T-1} \leftarrow n_0$
 12:     **while** $n_{T-1}$ is non-terminal **do**
 13:         **if** $n_{T-1}$ is not fully expanded **then**
 14:             **return** $n_{T-1}$
 15:         **else**
 16:             $n_{T-1} \leftarrow$ child node of $n_{T-1}$ with action sampling from Eqn. 13.
 17:         **end if**
 18:     **end while**
 19:     **return** $n_{T-1}$
 20: **end function**

 21: **function** EXPANSION$(n_{T-1})$
 22:     **if** $n_{T-1}$ is non-terminal **and** $T \neq H$ **then**
 23:         $n_T \leftarrow$ expand $n_{T-1}$ with an untried action.
 24:         **return** $n_T$
 25:     **else**
 26:         **return** $n_{T-1}$
 27:     **end if**
 28: **end function**

 29: **function** SIMULATION&EVALUATION$(n_T)$
 30:     **if** $n_T$ is non-terminal **and** $T \neq H$ **then**
 31:         $n_{\text{terminal}} \leftarrow$ Roll-out$(n_T)$
 32:     **else**
 33:         $n_{\text{terminal}} \leftarrow n_T$
 34:     **end if**
 35:     $\{s_T, a_T, \cdots, s_{\text{terminal}}, a_{\text{terminal}}\} \leftarrow$ extract from $n_{\text{terminal}}$.
 36:     $R \leftarrow \sum_{t=T}^{\text{terminal}} \gamma^{t-T} r_R(s_t, a_t)$
 37:     **return** $R$
 38: **end function**

 39: **function** BACK-PROPAGATION$(n_T, R)$
 40:     $t \leftarrow T - 1$
 41:     **while** $t \geq 0$ **do**
 42:         $Q_R(s_t, a_t) \leftarrow$ Eqn. 14
 43:         $N(s_t, a_t) \leftarrow N(s_t, a_t) + 1$
 44:         $t \leftarrow t - 1$
 45:     **end while**
 46: **end function**