# OpenReview forum: "Residual Q-Learning: Offline and Online Policy Customization without Value"
_NeurIPS.cc/2023/Conference — NeurIPS 2023 poster_

### Official Review · Reviewer_sftZ · 2023-07-02

**Soundness:** 2 fair
**Presentation:** 3 good
**Contribution:** 2 fair
**Rating:** 4
**Confidence:** 5

**Summary:**

This paper proposes a problem setting called policy customization, which involves training a policy that inherits the characteristics of a prior policy while satisfying additional requirements imposed by a target downstream task. They present a novel approach to interpreting and determining the two task objectives' trade-offs. Specifically, they formulate the customization problem as a Markov Decision Process (MDP) with a reward function that combines the inherent reward of the demonstration and the additional reward specified by the downstream task. The authors introduce a new framework called Residual Q-learning that leverages the prior policy to solve the formulated MDP without knowledge of the prior policy's inherent reward or value function. They derive a family of residual Q-learning algorithms capable of offline and online policy customization, demonstrating their effectiveness in accomplishing policy customization tasks in various environments.

**Strengths:**

1. The paper introduces a novel problem setting, policy customization, which addresses an important issue in imitation learning. This innovative approach better meets the personalized requirements of imitation policies in practical applications.

2. The paper provides experimental validation by conducting experiments in various environments, demonstrating the effectiveness of the proposed algorithms in achieving policy customization for real-world tasks. These experiments verify the practicality and adaptability of the methodology, providing strong support for further research and applications.

3. The proposed methodology in the paper is shown to be feasible and scalable. It does not rely on specific assumptions or constraints, making it applicable to different scenarios and tasks. The scalability of the approach allows for its extension and adaptation to more complex and diverse environments.

**Weaknesses:**

1. The derivation process in Equation 5 lacks clarity, particularly the transition from eq. 5b to eq. 5c and eq. 5d. It is necessary to provide a more detailed and step-by-step derivation to enhance the understanding of the readers. A comprehensive explanation of the intermediate steps and the mathematical transformations involved would greatly improve the transparency and rigor of the derivations.

2. While the proposed setting in the paper is intriguing, there is a need for further clarification on the rationale behind its adoption. It is unclear why the chosen measure is employed, and it would be beneficial to provide a clear justification for this decision. Additionally, if a prior policy is available, elaborating on how policy expansion can be utilized to address the problem posed in the paper would help readers grasp the underlying reasoning.

3. The experimental evaluation in this paper is limited in its scope. To substantiate the claims made, it is recommended to conduct more comprehensive experiments and provide extensive results. Expanding the experimental analysis with diverse scenarios, benchmark datasets, and evaluation metrics would contribute to a more thorough evaluation and provide stronger evidence to support the proposed approach.

4. The derived forms of Residual SQL and SAC in Eq. 7-10 appear to be similar to the original formulations. It is necessary to provide a clear explanation of how to interpret and understand the derived results. Highlighting the differences, if any, and elucidating the implications of the derived forms compared to the original ones would enhance the clarity and significance of the proposed modifications.

**Questions:**

Please refer to the Weakness.

**Limitations:**

Please refer to the Weakness.

---

> ### Author Rebuttal · Authors · 2023-08-08
>
> > The derivation process in Equation 5 lacks clarity, particularly the transition from eq. 5b to eq. 5c and eq. 5d. It is necessary to provide a more detailed and step-by-step derivation to enhance the understanding of the readers.
>
> Thank you for your suggestion on including a step-by-step derivation of Eqn. 5. In the revised manuscript, we plan to incorporate the following step-by-step breakdown of the derivation process in the appendix:
>
> First, we can write $r(s, a)$ as a function of the soft Q-function $Q^*(s, a)$ according to Eqn. (2):
>
> \begin{equation*}
>     r(s,a) = \gamma \mathbb{E}\_{s'\sim p(\cdot|s,a)}\left[\alpha \log \int_{\mathcal{A}}\exp\left(\frac{1}{\alpha}Q^* (s',a')\right)da'\right]-Q^*(s,a).
> \end{equation*}
>
> Eqn. (5b) results from substituting $r(s, a)$ with the right-hand side of the above equation. To derive Eqn. (5c), we first write $\int_{\mathcal{A}}\exp\left(\frac{1}{\alpha}Q^* (s',a')\right)da$ as the normalization factor $Z_{s'}$ based on its definition. Then note that, according to the definition of the residual Q-function $Q_{R,t}$, we can substitute $\hat{Q}\_t(s', a')$ in Eqn. (5b) with:
> \begin{equation*}
> \begin{aligned}
> \hat{Q}\_t(s',a') &= Q\_{R,t}(s', a')+\omega Q^*(s', a') \\\\
> & = Q\_{R,t}(s', a') + \omega \alpha \log \pi\left(a' \vert s'\right) + \omega\alpha \log Z\_{s'},
> \end{aligned}
> \end{equation*}
>
> where the second step comes from Eqn. (3). To obtain Eqn. (5d), note that $Z\_{s'}$ is not a function of $a'$, so we can write the last term of Eqn. (5c) as:
> \begin{equation*}
> \begin{aligned}
> &\mathbb{E}\_{s'}\left[\hat{\alpha}\log \int\_{\mathcal{A}}\exp\left(\frac{1}{\hat{\alpha}}\left({Q}\_{R,t} (s',a')+ \omega\alpha \log \pi\left(a' \vert s'\right) + \omega\alpha \log Z\_{s'}\right)\right)da'\right]\\\\
> = & \mathbb{E}\_{s'}\left[\hat{\alpha}\log Z\_{s'}^{\omega\alpha/\hat{\alpha}}\int\_{\mathcal{A}}\exp\left(\frac{1}{\hat{\alpha}}\left({Q}\_{R,t} (s',a')+\omega\alpha\log \pi\left(a' \vert s'\right)\right)\right)da'\right]\\\\
> = & \mathbb{E}\_{s'}\left[\hat{\alpha}\log \int\_{\mathcal{A}}\exp\left(\frac{1}{\hat{\alpha}}\left({Q}\_{R,t} (s',a')+\omega\alpha\log \pi\left(a' \vert s'\right)\right)\right)da'\right] + \omega \alpha \gamma \mathbb{E}\_{s'}\log Z_{s'}.
> \end{aligned}
> \end{equation*}
>
> > While the proposed setting in the paper is intriguing, there is a need for further clarification on the rationale behind its adoption. It is unclear why the chosen measure is employed, and it would be beneficial to provide a clear justification for this decision. Additionally, if a prior policy is available, elaborating on how policy expansion can be utilized to address the problem posed in the paper would help readers grasp the underlying reasoning.
>
> Could you clarify which parts of the paper you find unclear and need further clarification? In the author rebuttal thread, we further elaborated on the motivation of policy customization and its difference from RL fine-tuning and inverse RL. We hope it clarifies the rationale behind our proposed problem setting and solution to the readers.
>
> > The experimental evaluation in this paper is limited in its scope.
>
> We understand that the current experiments do not fully reflect the wide scope of applications our proposed theoretical framework could have. To address this issue, we have conducted additional experiments in MuJoCo environments. Please refer to the author rebuttal thread for a summary of the results. We plan to add these additional results in the revised version of the paper.
>
> > The derived forms of Residual SQL and SAC in Eq. 7-10 appear to be similar to the original formulations. It is necessary to provide a clear explanation of how to interpret and understand the derived results.
>
> Thank you for your suggestions. To make the derivations of the residual SQL and residual SAC clear to the readers, we plan to include a more detailed explanation in the appendix of the revised manuscript. Due to the space limit, we cannot post the complete derivation process here. We provide a sketch of the residual SQL's derivation for your reference. The residual SAC algorithm can be derived in a similar manner.
>
> The original objective of soft Q-learning is to minimize the TD error of the soft Q-function:
> $J\_{Q}(\theta) = \mathbb{E}\_{(s\_t,a_t)}\left[\frac{1}{2}\left(\hat{Q}^{\mathrm{target}}\_{\bar{\theta}}(s\_t,a_t) - \hat{Q}\_{\theta}(s\_t,a\_t)\right)^2\right]$,
> where the target Q-value is defined as $\hat{Q}\_{\bar{\theta}}^{\mathrm{target}}(s\_t,a\_t) = r\_{R}(s\_t,a\_t) + \omega r(s\_t,a\_t) + \gamma \mathbb{E}\_{s'\sim p(\cdot|s\_t,a\_t)}\left[\hat{\alpha}\log \int_{\mathcal{A}}\exp\left(\frac{1}{\hat{\alpha}}\hat{Q}_{\theta}(s',a')\right)da'\right]$.
>
> Given the defined residual Q-function, we can write the soft Q-function as
> $\hat{Q}\_{\theta}(s,a)=Q\_{R,\theta}(s, a) + \omega Q^*(s, a)$. Following steps similar to Eqn. (5), we can express the target Q-value with the residual Q-function as
> $\hat{Q}\_{\bar{\theta}}^{\mathrm{target}}(s\_t,a\_t) = r\_{R}(s\_t,a\_t) + \omega Q^*(s\_t,a\_t) + \gamma \mathbb{E}\_{s'\sim p(\cdot|s\_t,a\_t)}\left[\hat{\alpha}\log \int_{\mathcal{A}}\exp\left(\frac{1}{\hat{\alpha}}\left(Q\_{R,\theta}(s',a') + \omega'\log\pi(a'\vert s')\right)\right)da'\right].$
>
> Thus, if we define the target residual Q-value as $\hat{Q}\_{R,\bar{\theta}}^{\mathrm{target}}(s\_t,a\_t)=r\_{R}(s\_t,a\_t) + \gamma \mathbb{E}\_{s'\sim p(\cdot|s\_t,a\_t)}\left[\hat{\alpha}\log \int_{\mathcal{A}}\exp\left(\frac{1}{\hat{\alpha}}\left(Q\_{R,\theta}(s', a') + \omega'\log\pi(a'\vert s')\right)\right)da'\right]$, it is straightforward to see that the TD error of the soft Q-function equals to the TD error of the residual Q-function. It leads to the Eqn. (7-8) in the paper when we use the data in the replay buffer to estimate the target residual Q-value and the TD error.
>
> We hope our response can address your concerns and raise your impression of our work. Please let us know if you have any further questions.

---

> > ### Comment · Reviewer_sftZ · 2023-08-17
> > **Response**
> >
> > Thank you for your reply, which resolved some of my confusion. However, the entire manuscript still requires a significant amount of improvement. I have raised my score.

---

> > > ### Author Response · Authors · 2023-08-17
> > >
> > > Thank you for your feedback! We're pleased that our response has resolved some of your confusion. If you could kindly inform us about any lingering questions you have concerning our paper, we'd be delighted to provide additional clarifications. Thank you very much!

---

### Official Review · Reviewer_ePZC · 2023-07-04

**Soundness:** 3 good
**Presentation:** 4 excellent
**Contribution:** 3 good
**Rating:** 8
**Confidence:** 4

**Summary:**

This paper proposes a new problem setting called pulicy customization, where the goal is to train a policy that inherits the characteristics of a prior policy while satisfying some additional requirements imposed by a downstream task. To solve this problem, this paper uses a novel and principled framework by formulating it as a MDP with a reward function that combines the inherent reward of the demonstration and the add-on reward of the downstream task. The framework is consistent with many prior RL algorithms, and leads to residual Q-learning, residual SAC, and residual ME MCTS. A comprehensive evaluation of the proposed residual Q-learning framework is conducted on various types of environments, and algorithms with residual Q outperform the prior RL or IL policies in all of them.

**Strengths:**

- It introduces a novel and general problem setting of policy customization that can be applied to various domains and tasks.
- It proposes a principled and flexible framework of residual Q-learning that can customize a policy without knowing the reward or value function of the prior policy, which can be easily added to existing algorithms.
- Following the previous point, it provides a family of algorithms that can learn the residual Q-function and customize the policy.
- For experiments, it demonstrates the effectiveness and robustness of the residual Q-learning framework on both discrete and continuous tasks, showing that the customized policies can inherit the characteristics of the prior policy while satisfying the downstream task requirements.

**Weaknesses:**

While this paper has some vague weaknesses and limitations, they are fully discussed in Section 6 (Discussion and Limitations). So I would like to leave blank here.

**Questions:**

There is no direct questions but I am curious about the following points.
- Does the performance of residual-Q highly rely on the performance of the prior policy? For example, if the prior policy is learned from a mediocre dataset in an offline manner, will the residual Q-learning solve the following downstream task?
- What is the relationship between the policy customization problem and the continuous learning problem? Is the policy customization a special case of continuous RL or transfer RL?

**Limitations:**

Yes.

---

> ### Author Rebuttal · Authors · 2023-08-08
>
> Thank you for your valuable feedback. We are very excited that you find policy customization a novel problem setting and find the proposed residual Q-learning framework a principled and flexible solution to the policy customization problem.
>
> > Does the performance of residual-Q highly rely on the performance of the prior policy? For example, if the prior policy is learned from a mediocre dataset in an offline manner, will the residual Q-learning solve the following downstream task?
>
> This is a crucial point in the application of our approach, thank you for raising this question. Yes, the performance of residual Q-learning indeed relies on the performance of the prior policy. It is decided by the nature of the policy customization problem --- we rely on the prior policy to encode the inherent characteristics of the demonstration. In the submitted manuscript, we discussed this issue in Section 5.2 (lines 294-321, page 8). When we investigated residual Q-learning in the Parking environment, we actually started with an imperfect IL prior policy, since the environment is quite challenging given the continuous action space and, more importantly, the nature of the task—navigation in tight space under the non-holonomic constraint. We observed that policy customization improves the performance of the IL policy on the basic task. One aspect that we think can explain this phenomenon is that the add-on reward *implicitly* guides the policy to improve its basic task performance. Inspired by this observation, we think a practical solution to mitigate the issue of a mediocore prior is partially incorporating some basic task objectives into the add-on reward if available. It is reasonable for many practical use cases even with an imitative prior. While it is difficult to precisely describe the underlying reward function of human experts, we can comfortably assume that they follow some obvious commonsense objectives (e.g., collision avoidance). In this way, our method can be considered a more principled framework for the general synergy between IL and RL beyond policy customization where the reward functions are completely decomposed.
>
> > What is the relationship between the policy customization problem and the continuous learning problem? Is the policy customization a special case of continuous RL or transfer RL?
>
> Thank you for raising the connection between policy customization and continual learning. It is an excellent point and worthy of in-depth study in future work. The policy customization problem can be considered a special case of continual learning [1] with two stages. The similarities lie in 1) the agent learns to incrementally conquer the basic task and then the task with additional objectives; 2) the customized policy should retain its ability to fulfilling the basic task objective, i.e., avoiding catastrophic forgetting. In this sense, policy customization differs from transfer RL, since transfer RL primarily concerns the policy's performance on the transferred target task. In the revised manuscript, we plan to mention this connection at the end of Section 6 and list the in-depth study of policy customization under continual learning as a future work. In particular, an interesting direction is to study sequential policy customization (i.e., customizing the policy with multiple add-on rewards added sequentially), which could benefit from drawing insights from continual learning literature.
>
> [1] K. Khetarpal, et al. "Towards Continual Reinforcement Learning." *Journal of Artificial Intelligence Research (JAIR)*, 2022.

---

> > ### Comment · Reviewer_ePZC · 2023-08-14
> >
> > Thanks for your clear clarification! Now I think I have understood the details well.

---

### Official Review · Reviewer_QjWD · 2023-07-05

**Soundness:** 3 good
**Presentation:** 4 excellent
**Contribution:** 3 good
**Rating:** 7
**Confidence:** 4

**Summary:**

<< I have read the authors' rebuttal and have raised my score based on the discussion >>

This paper introduces a novel problem setting, named 'policy customization,' which seeks to train a new policy that inherits the properties of an existing pre-trained policy and meets additional requirements from a given downstream task. The policy customization is formulated as a Markov Decision Process (MDP), with a reward function that combines the inherent reward of the demonstration and an additional reward defined by the downstream task. To address the challenge of solving the MDP without knowledge of the underlying reward of the imitative policy, the paper proposes a novel Residual Q-Learning framework. This includes defining a residual Q-function that, when combined with the log-likelihood of the prior policy, can construct the maximum-entropy policy for the target MDP. The paper presents two model-free policy customization algorithms, namely, 'residual soft Q-learning' and 'residual soft actor-critic' and empirically demonstrates that the proposed residual Q-learning algorithms effectively customize the policies toward the additional task objective, while preserving their performance on the original tasks for which the prior policies were trained.

**Strengths:**

S1. Clarity of Presentation: The paper is commendably articulated, offering clear motivations for the research and a thorough explanation of the proposed solution. The technical details are presented in a digestible manner, which significantly enhances the reading experience.

S2. Technical Soundness: The paper exhibits robust technical soundness. The equations and underlying logic are laid out explicitly, providing a comprehensive understanding of the objectives. The problem formulation adheres to rigorous technical standards.

S3. Novelty: The paper contributes original research to a specific problem domain - policy customization - and I have not seen papers targeting this problem. As such, the solution proposed is novel.

Overall, the paper provides a compelling read and I found the research intriguing. However, some concerns about the experimental setup were noted (refer to the 'Weaknesses' section for details).

**Weaknesses:**

Despite the innovative approach presented in this paper, certain aspects of the experimental setup present potential limitations:

W1. The selection of relatively simple problems for testing constrains the perceived impact of the work. With more complex environments like Atari and Procgen games now serving as standard test suites, the study's relevance might be diminished due to the primitive nature of the chosen experimental domains.

W2. The paper lacks a comparison with a directly relevant baseline — RL fine-tuning combined with the IL objective of the prior policy used as heuristics. Given that RL or IL prior policies are trained on objectives distinct from the downstream task, their performance on the latter is likely to be sub-optimal. A head-to-head comparison with this method, which is a natural consideration for the policy customization problem, would better establish the strengths and limitations of the proposed Residual Q-learning framework.

**Questions:**

Q1. This work proves beneficial when the downstream task's objective is orthogonal to the prior policy's objective. However, what happens when the objectives conflict? For instance, suppose a prior policy for robot navigation is conservatively trained to minimize speed and avoid collision. If the downstream task necessitates rapid completion, wouldn't the prior policy potentially impede the optimization of the downstream task's objective?

Q2. If possible, could you share the results of experiments where the Residual Q-learning framework is compared with RL fine-tuning combined with the IL objective of the prior policy used as heuristics? Such comparisons would provide a more comprehensive evaluation of the proposed approach.

Q3. Based on the results, "RL with total reward" consistently outperforms the Residual Q-learning framework. Could you elucidate potential real-world scenarios where the Residual Q-learning framework would be more advantageous or preferred over the "RL with total reward" approach?

**Limitations:**

The authors discussed the limitations of their work. No discussion needed regarding potential negative societal impact.

---

> ### Author Rebuttal · Authors · 2023-08-07
>
> Thank you for your thoughtful feedback and suggestions. We are very excited that you find our paper commendably articulated, novel, and clearly motivated. We are also glad to have your recognition of the technical soundness of the problem formation and derivation. Please find below our responses to address your questions regarding our work:
>
> >The selection of relatively simple problems for testing constrains the perceived impact of the work.
>
> We understand that the current experiments do not fully reflect the wide scope of applications our proposed theoretical framework could have. To address this issue, we have conducted additional experiments in MuJoCo environments. Please refer to the author rebuttal thread for a summary of the additional experimental results. We plan to add these additional experiments in the revised version.
>
> >The paper lacks a comparison with a directly relevant baseline — RL fine-tuning combined with the IL objective of the prior policy used as heuristics.
>
> Thank you for proposing the RL fine-tuning baselines. They are indeed very important baselines for comparison. In the submitted manuscript, we have actually investigated two representative RL fine-tuning methods under the context of policy customization and discussed our findings in Section 6 (lines 329-361, pages 8-9) and Appendix D.1-2. Please refer to the "Comparison with RL Fine-tuning" paragraph in the author rebuttal thread for a brief summary.
>
> >This work proves beneficial when the downstream task's objective is orthogonal to the prior policy's objective. However, what happens when the objectives conflict? For instance, suppose a prior policy for robot navigation is conservatively trained to minimize speed and avoid collision. If the downstream task necessitates rapid completion, wouldn't the prior policy potentially impede the optimization of the downstream task's objective?
>
> Thank you for your insightful question. When the objectives of the prior policy and the downstream task conflict, our residual Q-learning framework still provides a principled way to interpret and determine the trade-off between these conflicting objectives. We would not consider the prior policy to impede the performance of residual Q-learning in this circumstance. Our goal is not to find the optimal policy that merely maximizes the add-on reward but achieves the desired trade-off between the basic (imitative) and downstream task objectives.
>
> >Could you elucidate potential real-world scenarios where the Residual Q-learning framework would be more advantageous or preferred over the "RL with total reward" approach?
>
> As noted in Section 5.1 (line 278, page 7), we do not intend to treat "RL with total reward" as a baseline for fair comparison under the policy customization setting, since it requires ground-truth knowledge of the basic reward. In policy customization, we are mainly interested in application domains where handcrafting the basic reward function is challenging, but large-scale demonstrations are available to synthesize imitative prior policies. We included the "RL with total reward" baseline for comparison with residual-Q to validate that the customized policies indeed solve the overall task $\hat{\mathcal{M}}$.
>
> We hope our clarifications and additional experimental results can address your concerns and raise your impression of our work. Please let us know if you have any further questions.

---

> > ### Comment · Reviewer_QjWD · 2023-08-14
> > **Thanks for addressing the concerns**
> >
> > ### Results on MuCoCo
> > it appears that the performance comparison between Greedy and Residual Q is quite close. Aside from the Ant environment, there doesn't seem to be a clear dominance of Residual Q over Greedy. Could the authors specify the value of the hyperparameter used for the IL objective?
> >
> > ### General comment
> > After reading the other reviews and the authors' rebuttals, I'd like to thank the authors for presenting their view in addressing the concerns. I concur with other reviewers that both the general motivation for policy customisation and the proposed method need better articulation, which would enhance the significance of Residual Q-learning. I've reviewed the experiments in appendix D.1-2 and believe they should be included in the main manuscript, as they represent key alternatives to Residual Q-learning for policy customisation. Residual Q-learning seems a more grounded approach to handling IL and downstream RL objectives. Nonetheless, I think the paper could benefit from broader and more complex experimental domains for comparison with the alternatives outlined in appendix D1-2, for clear or significant dominance over the alternatives. Due to these reasons, I'm maintaining my original score.

---

> > > ### Author Response · Authors · 2023-08-15
> > >
> > > Thank you for your feedback! We would like to elaborate further on what the experimental results of greedy vs. residual-Q policies implies. When compared to RL fine-tuning (i.e., RL/IL greedy), residual Q-learning consistently achieves better performance when a sufficiently good prior is available. The advantages are particularly apparent in the Parking and Ant environments. Note that the performance advantage is also apparent in the Humanoid environment when the prior policy is trained with RL ---- the greedily customized policy has significantly larger variances in its rewards compared to the residual Q-learning policy. The performance gap between residual-Q and RL fine-tuning depends on the basic and add-on rewards. If the additional add-on reward does not significantly alter the optimal policy, policy divergence is a fairly reasonable IL objective and it is fairly easier to find a good trade-off between the IL and RL objectives, even though policy divergence is a heuristically designed IL objective. For example, it is the case in the Hopper environment ---- while the customized agents jump higher, the gaits are still similar to the prior policy. In contrast, if the add-on reward does significantly alter the optimal policy, regularizing the policy KL divergence does not necessarily maintain the customized policy's performance on the basic task. For example, in the Parking environment, because of the non-holonomic constraint and long task horizon, the agent's optimal behavior varies significantly when the additional non-collision constraint is imposed. Thus, RL fine-tuning fails to find a good customized policy that can reliably complete the basic parking tasks. Conversely, a good prior policy can still be used to encode the basic task objective in the residual Q-learning framework, which explains the dominance of the residual-Q policies over the greedy ones. Therefore, even though the greedy policies perform similarly to residual-Q in some environments, the fact that residual-Q can consistently outperform RL fine-tuning given a good prior validates that residual-Q is a more principled approach to handling the trade-off between IL and downstream RL objectives.
> > >
> > > Regarding your question on the hyperparameters used for the IL objective, could you let us know which hyperparameters you refer to and are most curious about?

---

> > > > ### Comment · Reviewer_QjWD · 2023-08-19
> > > >
> > > > Thank you for addressing my concerns. In my initial review, I referred to the λ parameter in eq. 25 and the β parameter in eq. 28. It is my understanding that these parameters determine the balance between retaining information from the prior policy and optimising the add-on RL objective.

---

> > > > > ### Author Response · Authors · 2023-08-20
> > > > >
> > > > > In each environment tested, we set the value of $\lambda$ to be the same as the $\omega'$ value set in Residual Q-learning (see Table 4 in Appendix), so that Eqn. 27 is consistent with Eqn. 6 in terms of the weights of the prior policy likelihoods. Regarding $\beta$, we set it to be the same as $\omega'$ as well in the experiment reported in Fig. 2. Note that we have tested a wide range of $\beta$ values, and none of them gave satisfying results.

---

> > > > > > ### Comment · Reviewer_QjWD · 2023-08-20
> > > > > >
> > > > > > Thank you for clarifying. I will raise my score towards acceptance.

---

### Official Review · Reviewer_2HY5 · 2023-07-06

**Soundness:** 3 good
**Presentation:** 3 good
**Contribution:** 2 fair
**Rating:** 4
**Confidence:** 4

**Summary:**

This paper proposes a new problem setting called policy customization, where the goal is to train a new policy that inherits the characteristics of a prior policy while satisfying additional requirements imposed by a downstream task. The authors propose a novel Residual Q-Learning framework to solve this problem, which leverages the prior policy without knowing its inherent reward or value function. The authors derive a family of residual Q-learning algorithms for offline and online policy customization and show their effectiveness in various environments. The empirical results on four tested environments seem promising.


**Strengths:**

- The paper introduces a new problem setting called policy customization, which addresses the need to customize a pre-trained policy to meet additional requirements imposed by a downstream task.
- The Residual Q-Learning framework proposed in this paper provides a principled approach to interpret and design the policy customization objective, enabling the joint optimization of imitative and downstream task performance.


**Weaknesses:**

- The evaluation for the proposed method is limited. It would be beneficial to see more extensive comparisons and discussions with more baselines in the field, such as EGPO [1] and TS2C [2]. Also it would be beneficial to include some larger-scale RL environments, such as MuJoCo or MetaDrive [3].
- A more comprehensive discussion with related works should be presented. There is a whole line of research that assumes the availability of an online expert policy to imitate [1,2,4,5,6]. With these methods, it will be natural to perform policy customization by adding task-specific rewards to the learning process.

**Questions:**

- What is the advantage of Residual Q-Learning when compared with inverse-RL? One can learn the inherent reward function from interactions and combine it with the additional task-specific reward.
- What does it mean by “principled way”? Why is Residual Q-Learning better than related works?
- How is the prior policy $\pi$ used? Are you assuming you have an “online” policy to imitate? Then you may also need to compare with those papers focusing on RL with shared control [1,2,6].


**Limitations:**

References

[1] Peng, Zhenghao, et al. "Safe driving via expert guided policy optimization." Conference on Robot Learning. PMLR, 2022.

[2] Xue, Zhenghai, et al. "Guarded Policy Optimization with Imperfect Online Demonstrations." The Eleventh International Conference on Learning Representations. 2022.

[3] Li, Quanyi, et al. "Metadrive: Composing diverse driving scenarios for generalizable reinforcement learning." IEEE transactions on pattern analysis and machine intelligence 45.3 (2022): 3461-3475.

[4] Kelly, Michael, et al. "Hg-dagger: Interactive imitation learning with human experts." 2019 International Conference on Robotics and Automation (ICRA). IEEE, 2019.

[5] Mandlekar, Ajay, et al. "Human-in-the-loop imitation learning using remote teleoperation." arXiv preprint arXiv:2012.06733 (2020).

[6] Li, Quanyi, Zhenghao Peng, and Bolei Zhou. "Efficient learning of safe driving policy via human-ai copilot optimization." arXiv preprint arXiv:2202.10341 (2022).

---

> ### Author Rebuttal · Authors · 2023-08-09
>
> Thank you for your thoughtful feedback and valuable suggestions. We are glad you find the policy customization problem and residual Q-learning framework novel. Please find below our responses to address your concerns regarding our work:
>
> > The evaluation for the proposed method is limited. It would be beneficial to include some larger-scale RL environments, such as MuJoCo or MetaDrive.
>
> Thank you for suggesting additional experiments in MuJoCo environments. We understand that the current experiments do not fully reflect the wide scope of applications our proposed theoretical framework could have. To address this issue, we followed your suggestions and conducted additional experiments in MuJoCo environments. Please refer to the author rebuttal thread for a summary of the additional experimental results. We plan to add these additional experiments in the revised version of the paper.
>
> > What is the advantage of Residual Q-Learning when compared with inverse-RL?
>
> Thank you for raising inverse RL for comparison. It is indeed a very important point. Please refer to the author rebuttal thread for our clarification on this question. We plan to add the discussion of inverse RL in Section 2.2 of the revised version, in order to motivate our choice of customizing the policy without inferring its underlying reward.
>
> > What does it mean by “principled way”? Why is Residual Q-Learning better than related works?
>
> Policy customization aims to train a policy that inherits the characteristics of an imitative prior while satisfying additional downstream task requirements. Residual-Q is a principled solution to policy customization because it provides a theoretically sound way to interpret and determine the trade-off between the imitative objective and additional task requirements, especially compared to the widely studied RL fine-tuning methods. Many RL fine-tuning methods also combine IL and RL objectives. However, the IL objective is typically chosen as either divergence in policy distribution or distance in action space. While it is not necessarily an issue in RL fine-tuning problem -- the IL objective is added as a regularization to accelerate RL training -- it becomes an issue when we consider these methods in the context of policy customization. It is challenging to determine a good trade-off between two very different objectives which evaluate the policy in distinct dimensions. In contrast, in residual-Q, the trade-off between these two objectives is equivalent to the trade-off between the basic and add-on rewards; Thus, by optimizing the residual-Q objective, we are directly optimizing the policy's performance on the basic and add-on tasks given a desired weight between these two tasks. It is supported by our experiments with RL fine-tuning methods for policy customization. For a brief summary, please refer to the "Comparison with RL Fine-tuning" paragraph in the author rebuttal thread.
>
> > A more comprehensive discussion of related works should be presented. There is a whole line of research that assumes the availability of an online expert policy to imitate. With these methods, it will be natural to perform policy customization by adding task-specific rewards to the learning process.
>
> Thank you for bringing these related and interesting papers to our attention. First, we would like to clarify that residual-Q does not assume access to an online expert (e.g., human or trained policy) for the overall task (i.e., MDP with the weighted sum of the basic and add-on rewards as its reward). We assume access to a prior policy that accomplishes the basic task without the add-on reward. The goal is to customize the prior policy to complete both the basic and add-on tasks. In this sense, our work is different from the referred literature on learning from human intervention [1, 3, 4, 5] whose objective is to train a good imitative policy from human feedback without environment reward. However, we think those works could be potentially useful for training a good prior policy with better generalizability, when a human expert is available during online training. Such a generalizable prior could tackle the distributional shift issue we discussed in the paper. In the revised manuscript, we will include [1, 3, 4, 5] when discussing potential solutions to the distributional shift issue in the "Limitations" subsection (Line 367-372).
>
> Regarding [2], it is similar to ours from the perspective of having an imperfect expert -- the prior policy in residual-Q can be considered as an imperfect expert for the overall task -- and fusing knowledge extracted from this imperfect expert and RL exploration. The difference is [2] aims to train a policy towards a known reward; Thus, a value function can be trained to assert if the expert is reliable. In our case, since we rely on the prior policy to encode the basic task reward implicitly, we cannot use such a value function to govern the fusion between imitation and RL. In the revised manuscript, we will discuss [2] in Section 4, since it provides a different perspective on fusing imitation and RL.
>
> We hope our response can address your concerns and raise your impression of our work. Please let us know if you have any further questions.
>
> [1] Zhenghao Peng, Quanyi Li, Chunxiao Liu, and Bolei Zhou. "Safe driving via expert guided policy optimization." CoRL 2022.\
> [2] Zhenghai Xue, Zhenghao Peng, Quanyi Li, Zhihan Liu, and Bolei Zhou. "Guarded Policy Optimization with Imperfect Online Demonstrations." ICLR 2023.\
> [3] Michael Kelly, Chelsea Sidrane, Katherine Driggs-Campbell, and Mykel J. Kochenderfer. "Hg-dagger: Interactive imitation learning with human experts." ICRA 2019.\
> [4] Ajay Mandlekar, Danfei Xu, Roberto Martín-Martín, Yuke Zhu, Li Fei-Fei, Silvio Savarese. "Human-in-the-loop imitation learning using remote teleoperation." 2020.\
> [5] Quanyi Li, Zhenghao Peng, and Bolei Zhou. "Efficient learning of safe driving policy via human-ai copilot optimization." 2022.

---

> > ### Author Response · Authors · 2023-08-18
> >
> > Dear Reviewer,
> >
> > Thank you again for your valuable feedback! Please let us know if we have addressed your concerns. We would be happy to discuss any unresolved points.

---

> > ### Comment · Reviewer_2HY5 · 2023-08-19
> > **Response**
> >
> > Thank you for submitting your rebuttal. The authors have made a commendable effort to address and answer my concerns, which is greatly appreciated. I would also like to thank the authors for considering and discussing the related works. However, my other concerns remain as follows:
> > - Additional Experiments: The improvements of Residual-Q over baseline algorithms are extremely minor. I would expect more convicing empirical results as the novelty of the algorithm is relatively limited.
> > - Comparison with IL: The referred paper only demonstrate that IL shows worse performance than GAIL in certain environments. The authors may still need to discuss why Residual-Q, which require online policy queries, is a more preferred approach than IL.
> > - Principled way: The trade-off between different rewards can also be challenging, considering the whole line of research in multi-objective RL.
> >
> > Due to the aforementioned reasons, I remain my rating towards rejection.

---

> > > ### Author Response · Authors · 2023-08-20
> > >
> > > Thank you for your follow-up comments! Please find our responses to your remaining concerns as follows:
> > >
> > > **Improvement of Residual-Q over Baseline**: We respectfully disagree with your assessment that the improvements of Residual-Q over the RL fine-tuning baseline are extremely minor. When compared to RL fine-tuning, residual Q-learning consistently achieves better performance when a sufficiently good prior is available. The advantages are particularly apparent in the Parking and Ant environments. Note that the performance advantage is also apparent in the Humanoid environment when the prior policy is trained with RL ---- the greedily customized policy has significantly larger variances in its rewards compared to the residual Q-learning policy. The performance gap between residual-Q and RL fine-tuning depends on the basic and add-on rewards. If the additional add-on reward does not significantly alter the optimal policy, policy divergence is a fairly reasonable IL objective and it is fairly easier to find a good trade-off between the IL and RL objectives, even though policy divergence is a heuristically designed IL objective. For example, it is the case in the Hopper environment ---- while the customized agents jump higher, the gaits are still similar to the prior policy. In contrast, if the add-on reward does significantly alter the optimal policy, regularizing the policy KL divergence does not necessarily maintain the customized policy's performance on the basic task. For example, in the Parking environment, because of the non-holonomic constraint and long task horizon, the agent's optimal behavior varies significantly when the additional non-collision constraint is imposed. Thus, RL fine-tuning fails to find a good customized policy that can reliably complete the basic parking tasks. Conversely, a good prior policy can still be used to encode the basic task objective in the residual Q-learning framework, which explains the dominance of the residual-Q policies over the greedy ones. Therefore, even though the greedy policies perform similarly to residual-Q in some environments, the fact that residual-Q can consistently outperform RL fine-tuning given a good prior validates that residual-Q is a more principled approach to handling the trade-off between IL and downstream RL objectives.
> > >
> > > **Novelty**: The novelty of our work lies in both the new problem setting, policy customization, and the proposed solution, residual Q-learning. The residual-Q learning framework provides a theoretical ground to interpret and determine the trade-off between the imitative and add-on objectives, which has never been discussed in the related literature (e.g., RL fine-tuning). While the family of residual Q-learning algorithms is adapted from existing algorithms in a straightforward way, we consider the simplicity in algorithm design an advantage of our proposed residual-Q learning framework. As summarized by Reviewer ePZC, it shows that the proposed framework is flexible and can be easily added to existing maximum-entropy RL algorithms.
> > >
> > > **Comparison with IRL**: As mentioned in the author rebuttal thread, since the residual-Q framework does not require inferring the inherent reward from the demo, it allows flexible adoption of any maximum-entropy imitation learning algorithms depending on the environments, including AIRL. Thus, residual-Q should be able to consistently outperform policy customization methods that require explicitly inferring the inherent reward from the demo, since AIRL could perform significantly worse in certain environments. Besides, AIRL also needs to train a policy simultaneously while inferring the reward. Thus, we do not think the demand for a prior policy for online querying makes Residual-Q less favorable than IRL.
> > >
> > > **Principled Way**: We understand that determining the trade-off between different rewards can also be difficult in some scenarios. However, we want to emphasize that, compared to RL fine-tuning where the IL objective is added as a heuristic, interpreting the trade-off between IL and RL objectives as a trade-off in rewards as in Residual-Q is a more principled and theoretically sound approach. It has been validated by our experiments comparing Residual-Q versus RL fine-tuning.

---

### Official Review · Reviewer_6WQ5 · 2023-07-06

**Soundness:** 2 fair
**Presentation:** 2 fair
**Contribution:** 2 fair
**Rating:** 5
**Confidence:** 3

**Summary:**

This paper introduces a novel concept policy customization, which trains a new policy based on a pre-trained policy so as to solve the downstream tasks. Compared to RL fine-tuning algorithms, policy customization directly optimize the imitative objectives and downstream RL objectives jointly. The authors propose a new approach, Residual Q-learning, for policy customization. The proposed algorithm has shown effectiveness across 4 different environments in this work, suggesting their potential for efficient policy customization.

**Strengths:**

1. The paper tackles an important problem in RL, learning new tasks based on given policy priors.
2. The formulation and theorem are sound.
3. The approach is based on a simple concept "residual optimization", yet appears to generate some benefits, especially in harder tasks.

**Weaknesses:**

1. The author should clarify the significance of policy customization and make more arguments about the difference between policy customization and RL fine-tuning methods. The main difference between them seems that policy customization jointly trains the reward from the inherent rewards and add-on rewards, while RL fine-tuning methods learn from inherent rewards more intrinsically. Why is policy customization deserved to study, or why jointly combining the reward significant? Furthermore, I haven't noticed a detailed argument about the necessity of the proposed novel concept.
2. The experimental environments are simple. More complex tasks are better. Moreover, the authors should compare with some RL fine-tuning methods mentioned in related works.

**Questions:**

The optimization is based on the assumption that the optimal policy pre-trained with IL follows a Boltzmann distribution. The imitation policies in experiments are from stable-baselines' library. But for some more practical or complex envs, such as CARLA and MuJoCo, this assumption is hard to satisfy since the expert demonstrations have more and larger bias. I'm wondering the Eq (6) cannot be satisfied, how to optimize the Residual Q-Learning problems?

**Limitations:**

1. The statement about the definition and significance of policy customization is poor.
2. The experiment environments are simple, lacking convincing.
3. The assumption in Eq 1 is strong.

---

> ### Author Rebuttal · Authors · 2023-08-07
>
> Thank you for your thoughtful feedback and valuable suggestions. We are glad you find the policy customization problem novel and our theoretical formulation sound. Please find below our responses to address your concerns regarding our work:
>
> > The author should clarify the significance of policy customization and make more arguments about the difference between policy customization and RL fine-tuning methods. The main difference between them seems that policy customization jointly trains the reward from the inherent rewards and add-on rewards, while RL fine-tuning methods learn from inherent rewards more intrinsically. Why is policy customization deserved to study, or why jointly combining the reward significant? Furthermore, I haven't noticed a detailed argument about the necessity of the proposed novel concept.
> >
> Thank you for your suggestions on further elaborating on the significance of policy customization and its difference from RL fine-tuning. Please find our clarification on these questions and proposed revisions in the author rebuttal thread.
>
> > The experimental environments are simple. More complex tasks are better.
>
> We understand that the current experiments do not fully reflect the wide scope of applications our proposed theoretical framework could have. To address this issue, we have conducted additional experiments in MuJoCo environments. Please refer to the author rebuttal thread for a summary of the additional experimental results. We plan to add these additional experiments in the revised version of the paper.
>
> > Moreover, the authors should compare with some RL fine-tuning methods mentioned in related works.
>
> Thank you for proposing the RL fine-tuning baselines. They are indeed very important baselines for comparison. In the submitted manuscript, we have actually investigated two representative RL fine-tuning methods under the context of policy customization and discussed our findings in Section 6 (lines 329-361, pages 8-9) and Appendix D.1-2. Please refer to the "Comparison with RL Fine-tuning" paragraph in the author rebuttal thread for a brief summary.
>
> > The optimization is based on the assumption that the optimal policy pre-trained with IL follows a Boltzmann distribution. The imitation policies in experiments are from stable-baselines' library. But for some more practical or complex envs, such as CARLA and MuJoCo, this assumption is hard to satisfy since the expert demonstrations have more and larger bias. I'm wondering the Eq (6) cannot be satisfied, how to optimize the Residual Q-Learning problems?
>
> We understand the Boltzmann assumption is an important underlying assumption of our framework that need further justification. We consider the Boltzmann (i.e. maximum-entropy) assumption a reasonable one in imitation learning, since it has been commonly adopted to account for sub-optimal demonstration [1], which is particularly useful when the demonstration is collected from human experts. In the revised manuscript, we will add this reference to further justify the maximum-entropy assumption of demonstration when introducing Eqn. (1).
>
> Meanwhile, the demonstration's quality indeed affects residual Q-learning's performance. If the expert demonstration is limited and biased, then the IL prior cannot accurately represent the expert behavior. Since we rely on the IL prior to encode the inherent characteristics of the expert, the customized policy is not the optimal solution to the MDP with joint rewards due to the biased prior. However, this issue is not specific to policy customization and residual Q-learning but a fundamental question in learning from demonstration. Therefore, we believe it does not hinder the value of the newly proposed policy customization setting and the residual Q-learning framework. In the submitted manuscript, we discussed two potential solutions to mitigate this issue: 1) as discussed in Section 5.2 (lines 294-321, page 8), we can partially incorporate some basic task objectives into the add-on reward if available. It is reasonable for many practical use cases, even with an imitative prior. While it is difficult to describe the underlying reward function of human experts precisely, we can comfortably assume that they follow some obvious commonsense objectives (e.g., collision avoidance); 2) as mentioned in Section 6 (lines 369-372, page 9), we can adapt the recent advances in policy pretraining (e.g., [2]) to obtain a diverse imitative prior that is best suited for customization.
>
> [1] B. D. Ziebart, et al. "Maximum Entropy Inverse Reinforcement Learning." AAAI 2008.\
> [2] A. Singh, H. Liu, G. Zhou, A. Yu, N. Rhinehart, and S. Levine. "Parrot: Data-driven behavioral priors for reinforcement learning." ICLR 2021.
>
> We hope our clarifications and additional experimental results can address your concerns and raise your impression of our work. Please let us know if you have any further questions.

---

> > ### Author Response · Authors · 2023-08-18
> >
> > Dear Reviewer,
> >
> > Thank you again for your valuable feedback! Please let us know if we have addressed your concerns. We would be happy to discuss any unresolved points.

---

> ### Comment · Reviewer_6WQ5 · 2023-08-20
>
> Thank you for your reply and further experiments, which resolved some of my confusion and questions. I have raised my score.

---

### Author Rebuttal · Authors · 2023-08-06

We thank the reviewers for their thoughtful feedback and suggestions. We are delighted that the reviewers found policy customization novel (R1, R2, R3, R4, and R5) and essential (R1, R5). We are glad that they found residual-Q novel (R1, R2, R3, R4, and R5), principled (R2, R4), and flexible (R4). We also appreciate the reviewers for acknowledging the technical soundness of our derivations (R1, R3), and clarity of the technical details (R3). In this thread, we summarize our responses to the common concerns shared by the reviews. Please refer to the rebuttal attached to each reviewer's comments for a reviewer-specific response.

### Motivation
Several reviewers raised questions on the motivation for policy customization. Since it is a novel problem setting, we want to clarify these questions to help the reviewers assess the value of the proposed problem setting and solution. Policy customization aims to train a policy that inherits the characteristics of an imitative prior while satisfying downstream task requirements. It is useful for applications where: 1) handcrafting reward function is difficult, but demos are available to synthesize imitative policies; and 2) the downstream tasks impose diverse requirements beyond imitation. For example, autonomous vehicles should behave like human drivers to coordinate with other human-driven cars, which can be achieved through IL. Meanwhile, driving tasks impose additional objectives, such as reaching particular goals, enforcing safety, and complying with users' preferences. Residual-Q provides 1) a theoretical ground to interpret and determine the trade-off between the imitative and add-on objectives; and 2) flexible policy customization algorithms in offline and online manners.

**Comparison with RL Fine-tuning.** Multiple reviewers asked for clarification on the difference between residual-Q and RL fine-tuning. For clarification, we discussed this point in Sec. 4, Sec. 6, and Appx. D.1-2 of the submitted manuscript. Please find a summary of our arguments below. In the revised version, we will add the subtitle "Comparison with RL Fine-tuning" at the beginning of Sec. 6 to make this part more accessible to the readers.

In Sec. 4 (lines 194-201), we discussed the RL fine-tuning literature. In RL fine-tuning, the IL objective is added as a heuristic to accelerate RL training. Conversely, we aim to jointly optimize the IL and RL objectives in policy customization. In Sec. 6 (lines 329-361) and Appx. D.1-2, we investigated two representative RL fine-tuning methods under the context of policy customization, which are: 1) directly regularizing the KL-divergence between the trained RL and prior policies [1] during policy updates; and 2) adding the policy KL-divergence to the reward function [2, 3]. We found that both methods performed worse than residual-Q in the most challenging Parking environment --- RL fine-tuning failed to find a good trade-off between the imitative and add-on rewards. In contrast, residual-Q is able to find the policy that jointly optimizes the basic and add-on reward.

**Comparison with Inverse RL.** R2 asked for a comparison between inverse RL (IRL) and residual-Q. Indeed, learning the inherent reward via adversarial IRL (AIRL) [4] and then combining the inferred reward with task-specific rewards is a feasible solution to policy customization. However, AIRL could perform significantly worse than GAIL in certain environments [5]. Our residual-Q framework does not require inferring the inherent reward from demos; Thus, it allows flexible adoption of any maximum-entropy imitation learning algorithms depending on the environments (e.g., AIRL, GAIL, and IQ-learn [6]).

[1] A. Nair, et al. "Awac: Accelerating online reinforcement learning with offline datasets." 2020.\
[2] Y. Wu, et al. "Behavior regularized offline reinforcement learning." 2019.\
[3] M. Ziegler, et al. "Fine-tuning language models from human preferences." 2019.\
[4] J. Fu, et al. "Learning robust rewards with adversarial inverse reinforcement learning." ICLR 2018.\
[5] A. Gleave, et al. "Imitation: Clean imitation learning implementations." 2022.\
[6] D. Garg, et al. "Iq-learn: Inverse soft-q learning for imitation." NeurIPS 2021.

### Additional Experiments
We understand that the current experiments do not fully reflect the wide scope of applications our proposed theoretical framework could have. To address this issue, we conducted experiments in MuJoCo environments. The selected environments are Ant-v3, Humanoid-v3, and Hopper-v3. The basic task is to control the robot to move along the positive $x$ direction. During policy customization, we added an add-on reward to encourage the robot to also move along the positive $y$ direction for Ant and Humanoid. For Hopper, we added an add-on reward to encourage the robot to jump higher. The results are summarized in the attached file. In all the environments, the residual-Q customized policies achieved 1) higher add-on rewards than the prior policies; 2) a trade-off between the basic and add-on tasks similar to the RL full policies. In contrast, the RL fine-tuning baseline (i.e., the one described in Appx. D.1) tends to achieve higher add-on rewards but lower basic rewards compared to residual-Q and RL full policies. The total rewards of the RL fine-tuning baseline are also always lower than residual-Q. The results further validate that residual-Q outperforms RL fine-tuning in policy customization problems. The only exception is Humanoid with IL prior. Note that this IL prior was trained by BC since we did not find hyperparameters to let GAIL succeed. The BC prior is not an ideal prior suitable for residual-Q, as it does not follow the maximum-entropy policy distribution. It is reasonable that residual-Q performs worse than RL fine-tuning given the less diverse BC prior, since residual-Q relies on the prior policy to encode the basic task reward, whereas RL fine-tuning only uses the prior policy as a regularization.

---

### Decision · Program_Chairs · 2023-09-21

**Decision:**

Accept (poster)

**Comment:**

This paper addresses the problem of imitations learning in settings where an initial policy learned from demonstrations is intended to be adapted to a variety of downstream tasks. The authors formulate the problem as policy customization with several inferred reward components, and an approach called residual Q-learning designed to address the problem. The introduced approach is supported with theoretical insights and empirical validation.

Initial reviews saw a large difference in reviewer opinions. Noted strengths included the soundness of the approach as well as its generality, novelty and motivation of the problem formulation, demonstrated feasibility and scalability of the approach as shown in empirical validation, as well as a generally clear presentation. At the same time, concerns remained regarding the clarity and correctness of the presented theory, conceptual and empirical comparison with related work, such as fine-tuning for RL, as well as more general limitations of the empirical validation (baselines, tasks).

Reviewers noted that many concerns were resolved during the rebuttal period and assessments have become more positive overall. One remaining negative review initially noted an error in the theoretical part of the manuscript. The authors have resolved this issue in the rebuttal, and the relevant reviewer has indicated that their assessment has improved.

Overall, the assessment of the paper is now positive, with several reviewers strongly supporting acceptance. It is the ACs opinion that the clarifications, corrections and improvements discussed in the rebuttal period can be incorporated before the camera ready deadline. The recommendation is to accept, and the authors are strongly encouraged to take all feedback into account as they prepare the final version.